# Raman Spectroscopic Algorithms for Assessing Virulence in Oral Candidiasis: The Fight-or-Flight Response

**DOI:** 10.3390/ijms252111410

**Published:** 2024-10-24

**Authors:** Giuseppe Pezzotti, Tetsuya Adachi, Hayata Imamura, Saki Ikegami, Ryo Kitahara, Toshiro Yamamoto, Narisato Kanamura, Wenliang Zhu, Ken-ichi Ishibashi, Kazu Okuma, Osam Mazda, Aya Komori, Hitoshi Komatsuzawa, Koichi Makimura

**Affiliations:** 1Ceramic Physics Laboratory, Kyoto Institute of Technology, Sakyo-ku, Matsugasaki, Kyoto 606-8585, Japan; hyt8888@outlook.jp (H.I.); ikegami1771@gmail.com (S.I.); wlzhu@kit.ac.jp (W.Z.); 2Department of Molecular Genetics, Institute of Biomedical Science, Kansai Medical University, 2-5-1 Shinmachi, Hirakata, Osaka 573-1010, Japan; 3Department of Immunology, Graduate School of Medical Science, Kyoto Prefectural University of Medicine, Kamigyo-ku, 465 Kajii-cho, Kyoto 602-8566, Japan; t-adachi@koto.kpu-m.ac.jp (T.A.); mazda@koto.kpu-m.ac.jp (O.M.); 4Department of Dental Medicine, Graduate School of Medical Science, Kyoto Prefectural University of Medicine, Kamigyo-ku, Kyoto 602-8566, Japan; yamamoto@koto.kpu-m.ac.jp (T.Y.); kanamura@koto.kpu-m.ac.jp (N.K.); 5Department of Orthopedic Surgery, Tokyo Medical University, 6-7-1 Nishi-Shinjuku, Shinjuku-ku, Tokyo 160-0023, Japan; 6Department of Applied Science and Technology, Politecnico di Torino, Corso Duca degli Abruzzi 24, 10129 Torino, Italy; 7Department of Molecular Science and Nanosystems, Ca’ Foscari University of Venice, Via Torino 155, 30172 Venice, Italy; 8Department of Microbiology, Graduate School of Medicine, Kansai Medical University, 2-5-1 Shinmachi, Hirakata, Osaka 573-1010, Japan; okumak@hirakata.kmu.ac.jp; 9Department of Dentistry, Kyoto Prefectural Rehabilitation Hospital for Mentally and Physically Disabled, Naka Ashihara, Joyo, Kyoto 610-0113, Japan; 10Structural Biology Laboratory, College of Pharmaceutical Sciences, Ritsumeikan University, Shiga, Kusatsu 525-8577, Japan; ryo@ph.ritsumei.ac.jp; 11Laboratory of Host Defense and Responses, Faculty of Nutrition, Kagawa Nutrition University, 3-9-21 Chiyoda, Saitama, Sakado, Saitama 350-0288, Japan; ishibashi.kenichi@eiyo.ac.jp; 12Medical Mycology, Graduate School of Medicine, Teikyo University, Itabashi-ku, Tokyo 173-8605, Japan; kurodaya@med.teikyo-u.ac.jp (A.K.); makimura@med.teikyo-u.ac.jp (K.M.); 13Department of Bacteriology, Graduate School of Biomedical & Health Sciences, Hiroshima University, 1-2-3 Kasumi, Minami-ku, Hiroshima 734-8553, Japan; komatsuz@hiroshima-u.ac.jp; 14Teikyo University Institute of Medical Mycology (TIMM), 359 Otsuka, Hachijoji, Tokyo 192-0395, Japan

**Keywords:** oral candidiasis, Raman spectroscopy, virulence, morphogenic state, bedside analysis

## Abstract

This study aimed to test the effectiveness of Raman spectroscopy in the characterization of the degrees of physiological stress and virulence in clinical swab samples collected from patients affected by oral candidiasis. Raman experiments were conducted on a series of eight isolates, both in an as-collected state and after biofilm purification followed by 3 days of culture. The outputs were matched to optical microscopy observations and the results of conventional chromogenic medium assays. A statistically significant series of ten Raman spectra were collected for each clinical sample, and their averages were examined and interpreted as multiomic snapshots for *albicans* and non-*albicans* species. Spectroscopic analyses based on selected Raman parameters previously developed for standard *Candida* samples revealed an extreme structural complexity for all of the clinical samples, which arose from the concurrent presence of a variety of biofilms and commensal bacteria in the samples, as well as a number of other biochemical circumstances affecting the cells in their physiological stress state. However, three Raman algorithms survived such complexity, which enabled insightful classifications of *Candida* cells from clinical samples, in terms of their physiological stress and morphogenic state, membrane permeability, and virulence. These three characteristics, in turn, converged into a seemingly “fight or flight” response of the *Candida* cells. Although yet preliminary, the present study points out criticalities and proposes solutions regarding the potential utility of Raman spectroscopy in fast bedside analyses of surveillance samples.

## 1. Introduction

*Candida* yeasts, like many other microorganisms, endure a widely variable range of structural characteristics depending on the environmental conditions in their surroundings [1,2]. Nutrient availability, temperature, environmental pH, and oxygen level might induce different stress states in *Candida* clinical samples, which in turn, boldly affect the molecular composition of cells and lead to phenotypic changes that correlate with host specificity and mating [3]. Clinical *Candida* isolates are also exposed to different levels of immune response, antifungal agents, and treatment regimens, which affect their intrinsic characteristics and alter the clade population [4,5]. In other words, *Candida* samples retrieved from patients include isolates (not necessarily of a single species) with characteristics specific to the individual patient’s medical history, location, and previous exposure to healthcare settings [6]. In contrast, standard *Candida* strains are isolated and cultured long-term without exposure to external factors or stressors, so they exist in a conspicuously stress-free state. Accordingly, clinical samples display a unique mix of genomic and structural characteristics that generally do not match any single representative standard sample.

Beyond genomic speciation and specific clade/subclade classifications, Raman multiomic analyses of *Candida* samples directly reflect molecular structures and thus, if correctly interpreted, could give an instantaneous snapshot of various biological processes occurring at the sub-cellular level [7,8,9,10]. Raman analyses provide a holistic understanding of microorganisms’ biology, helping to rationalize the intrinsic characteristics of clinical samples, in terms of glycomics, lipidomics, proteomics, metabolomics, and other molecular traits. In particular, Raman spectroscopy of glycomics investigates the complex structure of carbohydrates and glycoproteins composing the *Candida* membrane [7]. Such knowledge is essential for understanding the role of cell surface glycans and their involvement in adhesion, recognition by the host immune system, drug resistance, and virulence [11]. Lipidomics can provide insights into how lipid composition and metabolism are associated with *Candida* pathogenicity, since lipids play crucial roles in membrane structure, signaling, and virulence in *Candida* species [12]. Raman spectroscopy could also provide insights into post-translational modifications and protein–protein interactions [13], as well as changes in metabolite levels and the related metabolic pathways [14,15]. Additional virulent molecules, such as candidalysin and other toxins secreted by pathogenic *Candida* species, could provide Raman fingerprints for morphogenesis and virulence [16]. However, multiomic analyses of *Candida* species are complex and require interdisciplinary collaboration among highly specialized researchers from a number of different fields. Note that a number of data mining procedures have been developed for the analysis of *Candida* species and the related biofilms, such as those based on mass spectrometry [17], nuclear magnetic resonance spectrometry [18], and other glycomic analyses [19,20]. These analytical methods can definitely provide a comprehensive understanding of *Candida* biology, pathogenesis, and host interactions. However, all those methods involve complex, expensive, and time-consuming procedures for both sample preparation and data collection. Conversely, once properly decrypted, Raman multiomic snapshots of *Candida* samples could, in a fast and inexpensive way, be similarly useful for understanding the structure of the fungal pathogen, its pattern of adaptation to the host environment, level of drug resistance, and virulence [7,8,9,10,21,22,23]. A Raman spectrum from a clinical *Candida* sample, once properly decrypted, can provide in a single snapshot a wealth of multiomic molecular information (including biofilms) [23,24] within minutes, namely, a time frame hardly achievable with any other analytical technique. For this reason, the Raman approach might be uniquely helpful in diagnostics, treatment strategies, and the development of new antifungal drugs, provided that advanced analytical algorithms and suitable data mining procedures are developed.

In this study, we apply Raman spectroscopy and imaging in order to discuss the differences between *Candida* samples from clinical patients affected by oral candidiasis and standard laboratory-grown (cultured) *Candida* samples, used as reference standards. Although the clinical samples examined here were all obtained from oral thrush recovered from patients not yet subjected to drug treatments, they exhibited widely altered molecular structures, as a consequence of being isolated from individuals with different backgrounds, immunologic characteristics, and health conditions. Accordingly, it appeared evident at the outset that clonal standard samples could not epitomize appropriate references because they could not represent the structural diversity of *Candida* pathogens from the clinical samples. Samples from patients generally consisted of a mixture of different biological clades and species, with cells embedded into the biofilms and extracellular matrices (i.e., including those from bacteria). With the aim of developing a Raman procedure capable of supporting chair-side analyses and effective drug selection, we attempted here to rationalize the differences between clinical and standard samples of oral *Candida* species beyond their general characterization into clades/subclades. Despite the complexity of such a task, some key Raman parameters could be located, which could be directly related to cells’ physiological stress/morphogenic state, membrane permeability, and virulence.

## 2. Results

### 2.1. Microscopy Observations and Clinical Characterizations

Figure 1a–p shows laser micrographs of clinical samples of oral candidiasis (cf. sample numbering in Table 1), as collected from patients and after purification–3-day culture, respectively. It was clear at the outset that there was an extreme difficulty in characterizing clinical samples by means of direct visual or morphological analyses, even after subjecting them to a purification–3-day-culture process. All samples appeared to be abundantly embedded into the extracellular matrix (ECM), which could not be completely eliminated upon a standard sample purification procedure. *Candida* ECM is known to play a number of different roles in the physiological behavior of yeast cells, including defense against phagocytosis and the obstruction of drug diffusion. However, the observed ECM was likely a composite structure, including biofilms from bacteria and a variety of biological phases, in addition to the possible presence of cells from a human host. The effect of the observed ECM on the Raman analyses of clinical samples will be discussed in more detail in successive sections.

Figure 2a–h shows the eight investigated clinical samples (cf. legends in inset; sample numbers are the same as Table 1) cultured for 3 days in BD BBL™ CHROMagar™ to identify species, and further cultured in Candida selective medium Candida detector (for an additional 3 days) to observe the respective purified colonies. This set of standard analyses, which was complemented with microscopy observation, revealed both greenish and brownish colorations, with a number of samples having mixed species composition, including *Candida albicans*, *Candida glabrata*, *Candida parapsilosis*, and *Candida tropicalis*, as listed in detail in Table 1.

### 2.2. Raman Spectroscopic Measurements

Figure 3a–h shows normalized Raman spectra averaged over sets of 10 spectra recorded on all as-collected and purified/3-day-cultured samples (cf. labels in inset). All Raman spectra were collected over the wavenumber interval of 300~1800 cm^−1^, which was divided into spectral zones I~VIII at increasing wavenumbers (cf. labels in inset). According to the speciation results shown in Figure 2, the labels *A*, *G*, *P*, and *T* in the inset identify the samples mainly consisting of *C. albicans*, *C. glabrata*, *C. parapsilosis*, and *C. tropicalis* species, respectively. The additional set of plots in Figure 3i–p (also divided into the same eight spectral zones as above) was obtained upon subtracting the spectra of purified/3-day-cultured samples from those of the respective as-collected samples. The encircled signs + and − (in inset) refer to the positive and negative spectral differences, respectively. As seen from these latter subtraction plots, all clinical samples experienced patterns different from each other, although such patterns also presented some common characteristics in specific spectral zones (e.g., Zones IV, V, and VIII). These similarities in the subtraction plots of Figure 3i–p will be discussed in detail in the following sections.

Figure 4 shows the procedure followed for Raman data treatment for two specific samples, one (*C. albicans*) and three (*C. glabrata* also including a minor fraction of *C. albicans*; cf. Table 1). The average Raman spectra include the spectral deconvolution and wavenumber labeling (in cm^−1^) of selected vibrational signals over the full investigated spectral interval 300~1800 cm^−1^. High-resolution laser micrographs of as-collected and purified/cultured Sample 1 are given in Figure 4a,b, respectively. The respective (average) Raman spectra (i.e., the same as those shown in Figure 3a) are shown in Figure 4c,d (cf. labels in inset) after deconvolution into Gaussian–Lorentzian sub-bands. The subtraction plot between the two spectra (i.e., the same as that shown in Figure 3i) is also included between Figure 4c,d for easier visualization. Similar characterizations are reported for Sample 3 (*C. glabrata*), for which high-resolution laser micrographs of as-collected and purified/3-day-cultured samples are shown in Figure 4e,f, respectively. The corresponding average and deconvoluted Raman spectra are given in Figure 4g,h (the same as shown in Figure 3c), together with the corresponding subtraction plot (the same as that shown in Figure 3k). Samples 1 and 3 basically included the same sequence of spectral sub-bands, although the relative Raman intensities were clearly different. Note, in addition, that signals recorded in the same sample before and after purification–culture were also markedly different. Details of spectral deconvolution and differences in relative intensity for selected Raman signals will be discussed later.

### 2.3. Principal Component Analysis on Reference and Clinical Samples

Figure 5a–d shows DIC micrographs of standard samples of *C. albicans* LSEM865, *C. glabrata* LSEM47, *C. parapsilosis* LSEM868, and *C. tropicalis* LSEM1823, respectively. The cell size of all standard species was between 4~7 μm in average diameter. Despite similarities in cell morphology, visual and phenomenological analyses can provide some qualitative indication of clade speciation. Raman spectra for the above standard samples (shown in Figure 5e–h), respectively; cf. labels in inset) [7], despite clear similarities, also showed a number of different features. Accordingly, it was possible to refine vibrational analyses through PCA to an extent sufficient to enable a univocal determination of these four different Candida species at the level of standard samples, as already suggested in a previous work on oral Candida species [7]. Figure 5i,j shows correlation and covariance matrices, respectively, for PC2/PC1 loading vectors with reference to the entire spectral region of 300~1800 cm^−1^ [7]. As seen, PCA allowed for distinguishing among standard samples upon judging from the mere morphology of their Raman spectra. However, when we attempted to apply the same PCA analytical procedure to clinical samples, we came across a complete failure in the identification of the *Candida* species, not only in the as-collected samples but also in the purified/3-day-cultured ones. Plots for all as-collected and purified/cultured clinical samples are given in Figure 6a,b, respectively. Three different combinations of PCA loading vectors are shown with respect to both correlation and covariance matrices (cf. labels in inset). All plots refer to the entire spectral region of 300~1800 cm^−1^. Unlike the case of the reference strains (cf. Figure 5i,j), none of the PCA plots, either for the as-collected or purified/cultured samples, allowed speciation among the clinical samples from Raman spectral morphology. The PCA analyses in Figure 6 stigmatize the extreme complexity of clinical samples as compared with those collected on reference clades and call for a different approach in Raman analyses.

## 3. Discussion

### 3.1. The Multifactorial Origin of Raman Complexity in Clinical Samples

Standard reference strains of *Candida* are often used in basic microbiology research [25,26]. However, the present study demonstrated that reference strains, while providing comparative genomic references and eventually revealing the presence of key genes in drug resistance [27], cannot capture the structural diversity and variability observed in clinical samples from oral candidiasis patients. As a practical consequence of such complexity, unlike the Raman spectra collected on standard clades [7,8,9,10], the spectra from the clinical samples cannot be simply classified according to conventional data statistics (e.g., PCA analyses; cf. Figure 5 and Figure 6).

Several factors contribute to producing differences between *Candida* samples from patients and reference clades. First of all, the *Candida* population in swab samples is likely comprehensive of more than one species, although one is generally dominant in candidiasis pathogenicity. Moreover, the patient’s microbiota, which is influenced by various host-specific factors such as age, genetics, immune system, and overall health state, plays a key role. Each individual possesses a unique oral microbial community, whose composition (and the composition of the related biofilms) strongly reflects in the overall recorded Raman spectrum. Accordingly, the circumstance that swab samples from different patients present the same microbial composition is highly improbable. In this same context, one should also consider that the results of Raman analyses on oral samples cannot directly be applied to other forms of candidiasis because the conditions in the oral cavity greatly differ from those in the gastrointestinal tract or on the skin. Nevertheless, one could search for intrinsic Raman parameters that describe how *Candida* cells adapt and thrive when embedded into specific microenvironments. Analytical approaches could be conceived, for example, to describe differences in nucleic acid and protein composition between yeast and hyphal forms, which generally relate to a different nucleus-to-cytoplasm ratio, with morphogenic switching also impacting upon membrane carbohydrates. In individuals with weakened immune systems, the presence of an infection could alter the microbial composition at the infection site, leading to more structural differences compared to non-infected individuals. Moreover, the presence of biofilms from oral bacteria in non-purified clinical samples strongly affects the Raman spectrum, with signal components peculiar to the specific oral flora of the patient, which is the case in the present experiments (cf. Table 1).

In line with and as a consequence of the above reasoning, the present study demonstrated that Raman assessments of clinical swab samples retrieved from patients affected by oral candidiasis should rather be addressed to understand multiomic characteristics related to *Candida* cells’ stress state, morphogenic switch/virulence, and membrane permeability patterns than attempting to directly link the sample to standard reference strains. We shall hereafter demonstrate that, despite the high structural complexity of swab samples and *Candida* clinical isolates, Raman spectroscopy could yet provide spectroscopic algorithms capable of shedding light on key infection characteristics.

### 3.2. Raman Fingerprints for Oxidation of Sulfur-Containing Amino Acids

Yeast cells of the *Candida* genus were reported to undergo oxidation of sulfur-containing amino acids, namely, cysteine and methionine, under conditions of oxidative stress [28]. These amino acids contain sulfur in their side chains, and their oxidation leads to the production of various oxidized sulfur species [29]. Cysteine, in particular, is exposed to the environment and is highly susceptible to oxidation. The thiol group (–SH) in cysteine could be oxidized and successively form disulfide bonds (–S–S–) or stem as acidic sulfur compounds [30]. Methionine, once unburied from the protein structure, could also undergo oxidative modifications, leading to the formation of various products, including methionine sulfoxides and dehydroxymethionine [31,32]. In *Candida* yeasts, oxidative stress occurs when an imbalance persists between the production of reactive oxygen species (ROS) and cells’ ability to detoxify them or repair their resulting damage. ROS react with cellular components containing sulfur-containing amino acids, leading to their oxidation [33]. The response of yeast cells to oxidative stress involves the activation of various defense mechanisms, including antioxidant systems and repair pathways, to mitigate the damage caused by oxidation. In a context of quite dynamically evolving structural characteristics as a function of environmental conditions, analyses of Raman signals related to S–S and C–S bonds (i.e., intrinsically strong because of the concurrent circumstances of light atoms linked by highly covalent bonds) have, in principle, the potential of finely unfolding oxidative stress patterns in yeast cells, thus providing unique insight into the cellular response to environmental triggers.

Following the spectroscopic approach described in detail in previous papers [24,34,35], we monitored the signal triplet at 507, 523, and 545 cm^−1^ (in spectral Zone II), which arises from disulfide (chain) S–S bonds in different cysteine complex conformations [36,37]. On the other hand, the relatively strong signal at 483 cm^−1^ can be assumed to mainly originate from S–SH stretching [37,38]. Depending on the host environment (e.g., pH), the Raman intensity ratio of signals related to S–S vs. S–SH bonds, namely, the ratio, *R_S-S_* = (*I*_507_ + *I*_523_ + *I*_545_)/*I*_483_, can give a measure of the environmental effect on clinical isolates. Since fungal species usually establish an acidic environment, the *R_S-S_* value is expected to be low in standard samples, while any increase in the *R_S-S_* ratio, compared to the reference clade sample, should link to a molecular-level reaction (i.e., a state of stress) in the isolate structure in response to ROS interactions and/or to an increasingly alkaline environment. In the same spectral interval, the intensity of the C–C=O in-plane bending signal (at ~606 cm^−1^) from phenylalanine could be taken as representative of the number of living *Candida* cells scrutinized by the laser probe [39].

Figure 7a shows a schematic draft of the oxidative modification of cysteine residues by oxygen radicals, which leads to the formation of sulfenic acid and, upon reaction with a neighboring cysteine molecule, yields disulfide bonds [40]. In sections b-e of the same figure, S–S bond-related sub-bands in spectral Zone II are analyzed for standard *C. albicans* LSEM865, *C. glabrata* LSEM47, *C. parapsilosis* LSEM868, and *C. tropicalis* LSEM1823, respectively (i.e., as extracted from the deconvoluted spectra in Figure 5d,e, respectively). S–S and S–SH-related sub-bands and the computed *R_S-S_* ratios (cf. values in inset) are shown for all clinical samples (cf. labels) in Figure 7f,g for the as-collected and purified/3-day-cultured samples, respectively. The *R_S-S_* spectroscopic ratio represents cysteine structural degenerations, with S-related signals epitomizing fingerprints of environmental pH. The appearance of any signal enhancement at ~850 cm^−1^ could also be related to O–O bonds that are traceable to peroxydisulfate and thiosulfate ions [41]. This latter band, which lies within Zone IV and is also reported in Figure 7b–g, could be interpreted as an additional Raman fingerprint of environmentally driven alterations of S-containing amino acid molecules.

The *R_S-S_* ratio in all reference clades (Figure 7b–e), despite the differences in rotameric cysteine conformations, was consistently detected at the relatively low value of 1.3, as expected in a well-stabilized culture environment. This trend was in clear contrast with the heterogeneity recorded for the clinical swab samples. As a general trend for the *R_S-S_* ratio, all tested clinical samples after purification and 3-day culture showed quite uniform values in the relatively narrow interval of 1.5~2.0 (cf. Figure 7g). These values were generally lower than the respective values in the as-collected state (cf. Figure 7f) and always intermediate with respect to those of the reference samples (i.e., constant at 1.3 for all clades; cf. Figure 7b–e). Two exceptional cases were Samples 7 and 8, for which extremely high *R_S-S_* ratios (i.e., 11.9 and 10.2, respectively) were recorded in the as-collected state (cf. Figure 7f). Both these disproportionally high *R_S-S_* ratios, which are discussed later, arose from an exceptionally low intensity for the 483 cm^−1^ S–SH stretching band (cf. Figure 7f). As expected, however, the purification–culture procedure re-established a sulfur chemistry closer to that of standard samples. This is obviously a consequence of adding the nearly neutral CHROMagar pH (~i.e., ~6.1) used for cell culturing with the *Candida* cells after purification yet in a transition to fully recover their acidic environment and/or their original protein structure [42]. Accordingly, the exceptionally high *R_S-S_* ratios in the as-received Samples 7 and 8 became comparable to those of the other samples after purification–culture. The S–S-related features for all samples can be compared at a glance by observing Zone I in Figure 3i–p. A scenario could be expected, in which the observed sulfur chemistry is strongly affected by the concurrent presence of bacteria and related biofilms (i.e., all candidiasis samples indeed included *Streptococci*; cf. Table 1). A balance between environmental pH conditions and ROS concentrations within the biofilm in the presence of bacteria should set the observed *R_S-S_* ratios to a variety of different values. For example, an extremely toxic ROS environment accompanied by relatively high environmental alkalinity leads to extremely high *R_S-S_* values (i.e., as observed in Samples 7 and 8). Note, however, that also the immune reaction of the human body generates ROS in the presence of H_2_O_2_-producing bacteria in both the mouth and the gut. Moreover, commensal bacteria, such as *Enterococcus faecalis*, *Streptococcus sanguinis*, and *Lactobacillus* species, are known to secrete ROS into their surroundings, causing an inhibitory effect on the proliferation of *Candida* cells [43,44,45]. In an elegant demonstration of ROS interaction between *Candida* cells and commensal bacteria, Cruz et al. [46] used a *Caenorhabditis elegans* model of polymicrobial infection to show how the ROS activity of *Enterococcus faecalis* could generate a toxic environment that induces oxidative stress in *C. albicans* and, accordingly, reduces its virulence. In other words, protein-ROS reactions give rise to protein–protein cross links, oxidation of the peptide backbone, and oxidation of amino acid side chains, thus altering protein functions and activating stress-sensing proteins through the oxidation of cysteine residues, ultimately inducing programmed cell death in fungal pathogens upon directly interacting with nucleic acids and causing irreversible damage [47,48]. This latter circumstance could justify the relatively large intensity fluctuations detected for the phenylalanine band at 606 cm^−1^, a sensor for the relative fraction of cells in the probe (cf. the highly scattered C–C=O in-plane bending signal intensity in both Figure 7f,g). It, thus, appears that clinical samples of *Candida* cells, even after purification and 3-day culture, could neither fully recover the sulfur chemistry of their long-term-cultured state nor their apoptotic behavior (cf. Figure 7g). This latter observation is also supported by the observation that the O–O band at ~850 cm^−1^ from peroxydisulfate and thiosulfate ions remained significantly pronounced even after purification–culture (Figure 7g) compared to fully stabilized standard samples (cf. Figure 7b–e).

In summary, although giving useful insights into the sulfur chemistry of clinical samples (including biofilms), the *R_S-S_* ratio (as representative of the spectral Zone I) is strongly affected by a number of concurrent environmental circumstances. Accordingly, unlike the case of reference *Candida* cultures, it can hardly serve as a direct sensor for clade/subclade speciation or as a measure of the physiological stress state of *Candida* cells in clinical samples.

### 3.3. Oxidative Stress Assessments by Monitoring Cytochrome C Redox State

From a molecular physiology viewpoint, an additional effect of oxidative stress and oxidative imbalance in *Candida* cells is the redox reaction of cytochrome *c*. [49]. Cytochrome *c* is a small heme protein consisting of 104 amino acids [50], whose structure is highly pH-sensitive and switches from an interconvertible reduced form (haem Fe^2+^) to an oxidized one (haem Fe^3+^) [51]. Characteristic peaks of reduced cytochrome *c* can be found at 750 (pyrrole ring breathing mode), 1128 (C–CH_3_ stretching), 1313 (CH_3_ and CH_2_ twisting), and 1585 (C–C and C–CH stretching) cm^−1^, while the signal at 1638 cm^−1^ (C=C and C=N stretching) is only peculiar to the oxidized form [28,52,53,54,55]. As explained in previous papers [7,15], the bands at 1128, and 1585 cm^−1^ strongly overlap with signals from a number of different biomolecules (e.g., C–C skeletal stretching and CH_3_/CH_2_ bending). Accordingly, they could hardly serve as markers for cytochrome *c*. Pyrrole rings are also present in a variety of biomolecules. However, using a 532 nm (green) laser irradiation, resonance Raman conditions allow for enhancing the signals from hemes of type *b* and *c* in mitochondrial cytochromes above those of proteins [56]. Although intensity alterations of the band at 1313 cm^−1^ could be considered as mainly related to the status of cytochrome *c*, contributions from proteins and glucans cannot be completely ruled out. According to the above notions, we monitored variations in the ~750 and 1638 cm^−1^ Raman bands with respect to standard samples of the same clades in order to establish a spectroscopic path to evaluate and compare the physiological stress state of *Candida* clinical samples before and after purification–3-day culture. In particular, we computed the Raman intensity ratio, *R_Cyt_* = *I*_1638_/(*I*_750_ + *I*_1638_), as a semi-quantitative parameter for assessing the level of cytochrome *c* oxidation and named it the *cytochrome chemistry index*. Figure 8a shows the molecular structures of reduced and oxidized cytochrome *c* with the respective Raman vibrational modes used for stress assessments in *Candida* cells. Sections b–e of the same figure show sub-bands for the reduced and oxidized heme molecules (at 750 and 1638 cm^−1^, respectively) and the related *R_Cyt_* indexes (in inset) for standard *C. albicans* LSEM865, *C. glabrata* LSEM47, *C. parapsilosis* LSEM868, and *C. tropicalis* LSEM1823, respectively (i.e., as extracted from the deconvoluted spectra in Figure 5d,e, respectively). According to their near-zero *R_Cyt_* indexes (≤0.08 in all cases), all standard cultures experienced, as expected, a nearly fully reduced heme structure, namely, an almost completely stress-free metabolic state as a result of long-term culture. Unlike the physiological stress-free state of the standard samples, all clinical samples in the as-collected state experienced an *R_Cyt_* index close to 1 (i.e., between ~0.6 and 0.9) as proof of a highly oxidized state of their heme structures (Figure 8f). Looking at the representative (averaged) Raman spectra after purification and 3-day culture (Figure 8g), it appears immediately clear that the oxidative stress-induced changes in the redox state of cytochrome *c* in all *Candida* samples were significantly relaxed. This was clearly a consequence of cleaning up the sample from bacteria and giving some time to the cells to approach their physiological equilibrium. Nevertheless, the oxidative state of the cells yet appeared far from being fully released (cf. *R_Cyt_ =* 0.3~0.6). The present data are consistent with previous Raman studies of the redox state change of cytochrome *c* upon the metabolic stress in *Candida* cells [28,57,58].

In summary, the present data confirmed that a reduced state of cytochrome *c* in physiologically unstressed *Candida* cells turns into an oxidized one upon environmentally driven oxidative stress. The turnover of the redox state, whose reversibility was confirmed in this study, could reliably be monitored by Raman spectroscopy, thus providing a clear spectroscopic parameter for the in situ probing of the cells’ stress state and mitochondrial dysfunction [28,49,57,58].

### 3.4. Raman Characterization of Glucan Structures in Biofilm and Cell Walls

As discussed in Section 3.2, and also phenomenologically arguable from the pronounced morphological diversity of spectra in Figure 3a–h, the samples collected from oral candidiasis patients consisted of widely variable mixtures of different biological phases and biofilms as likely built up not only by *Candida* cells but also by commensal bacteria. Moreover, the ultimate molecular structure of the ECM is affected by extrinsic factors, such as the patient’s immunological status and oral hygiene (in the present study, we shall exclude the effect of drugs, since only patients not yet undergoing drug treatments were selected). It is known that *Candida* cells might dynamically develop different cell wall structures depending on their morphogenic state and in response to specific environmental conditions [59]. From a morphological viewpoint, the ECM observed in all as-collected samples (Figure 1a–h) corresponded to a general description given for *Candida* species, namely a “polymeric gel-like hydrated three-dimensional structure in which the cells become partially immobilized” [60]. The ECM is key in supporting the overall cells’ physiological behavior and, despite being both species- and environment-dependent, generally includes polysaccharides (glucans and mannans), monosaccharides (including) hexosamines, and proteins, including their glycosylated counterparts [61]. In analyzing ECMs and cell walls in *Candida* samples, we relied on the extensive review of vibrational modes in carbohydrates and polysaccharides published by Wiercigroch et al. [62] This review served as a spectral library in the assignment of the origin of vibrational bands. According to this spectroscopic approach, we first examined the Raman spectrum of biofilm isolated from a reference *C. albicans* LSEM865 (yeast) culture (cf. normalized and deconvoluted spectrum in Figure 9a), in order to compare it with the spectra from samples retrieved from candidiasis patients before and after a purification–3-day-culture process (cf. Figure 3a–h). Raman spectroscopic fingerprints for β–glucans (mainly β–1,3-linked chains with β–1,6-linked branches), α–1,6–glucans, and α–1,3–glucans could be found at 890, 919, and ~941 cm^−1^, respectively. All signals arise from C–O–C glycosidic linkage stretching vibrations (Figure 9b), but differ in wavenumber because of the different bond types and structural contexts in which they are embedded [63,64]. By exploiting such vibrational differences, two glucan-related Raman biochemical parameters, *R_G__β/α_* = *I*_890_/(*I*_890_ + *I*_919_ + *I*_941_) and *R_G_*_α3/α6_ = *I*_941_/(*I*_919_ + *I*_941_), could be located, which relate to the fractions of α– vs. *β*–glucans and α–1,3– vs. α–1,6-linked glucans, respectively. Note that the α–1,3-linked glucan structure is by far the most rigid and insoluble molecule among polysaccharides. On the other hand, the α–1,6-linked glucan structure is water soluble, and the β–glucan one is pliable and conspicuously hydrated [64,65,66,67]. Accordingly, Raman analyses of glucans (either belonging to biofilm or to cell walls) in *Candida* samples can, in principle, provide unique insight into the rigidity, hydrophobicity, and consequent drug penetrability of clinical candidiasis samples (independent of clade and sub-clade classifications). The parameters *R_G__β/α_* and *R_G_*_α3/α6_, as computed for standard *C. albicans* biofilm, are given in the inset of Figure 9a. According to a similar analysis, Figure 9c–f gives the selected C–O–C glycosidic bond-stretching sub-bands and the related *R_G__β/α_* and *R_G_*_α3/α6_ parameters (in inset) for standard *C. albicans* LSEM865, *C. glabrata* LSEM47, *C. parapsilosis* LSEM868, and *C. tropicalis* LSEM1823, respectively (i.e., as extracted from the deconvoluted spectra in Figure 5d,e, respectively). As seen from Figure 9a, the biofilm collected from the reference *C. albicans* LSEM865 sample preponderantly (~70 vol.%) consisted of β–glucans (*R_G__β/α_* = 0.67), with the ~30 vol.% fraction of α–glucans being mainly of the α–1,6-linked type (*R_G_*_α3/α6_ = 0.30) [7]. In other words, the content of α–1,3-linked glucans in the biofilm of *C. albicans* yeast was as low as ~9 vol.%. The structural polysaccharide in the yeast cell walls of all reference *Candida* samples (lacking biofilm) included 15~50 vol.% fractions of β-linked glucans, with fractions of α–1,3-linked polymorph on the order of 16~34 vol.% of the overall glucan content (cf. *R_G__β/α_* and *R_G_*_α3/α6_ parameters in the inset to Figure 9c–f, respectively). These Raman data prove the substantial difference in the glucans’ structure existing between the ECM and the cell walls in *Candida* cultures. 

They also suggest that the main fraction of α–1,3-linked glucans detected in the clinical samples should belong to cell walls. Glucans-related sub-bands are reported in Figure 9g,h (cf. also *R_G__β/α_* and *R_G_*_α3/α6_ ratios computed in the inset for all clinical samples) for as-collected and purified/3-day-cultured conditions, respectively. In the as-collected samples (Figure 9g), both spectroscopic ratios represent fundamental structural traits that cumulatively epitomize the biological characteristics of glucans in both the biofilm and cell walls, although, as proven above, the detected fractions of α–1,3-linked glucans could mainly be assigned to cell walls. Assessments of glucans in clinical samples can also give a vivid snapshot of the wide extent to which *Candida* structures are dynamically variable. In particular, the ability of *Candida* species to synthesize higher fractions of water-insoluble glucans (i.e., glucans rich in α–1,3–glycosidic linkages) in their cell walls is crucial in determining the structural and dynamical attributes of the yeast membrane.

The general trend for all of the as-collected samples pointed to a clear enhancement in the content of β–glucans as compared to standard samples (cf. Figure 9c–f). These results can easily be explained by considering the observed higher presence of biofilm in the probed samples (cf. Figure 1a–h), which, as seen in Figure 9a, is rich in β–glucans. This trend was substantially, but not completely, reduced after purification–3-day culture in the majority of the samples (i.e., compare *R_Gβ/α_* data in Figure 9h with the respective values for standard cultures), which might suggest that the treatment of sample purification has not completely succeeded in eliminating ECM β–glucan molecules from the clinical samples. Regarding the relative content of α–1,3–glycosidic linkages, independent of species, it was generally observed to be conspicuously higher in the as-collected samples (cf. Figure 9g) than in the corresponding standard samples (Figure 9c–f). Such an alteration, which was maintained after the cycle of purification–3-day culture (cf. Figure 9h), might indicate that *Candida* cells in clinical samples tend to develop a higher fraction of α–1,3–linkages in their cell walls compared to standard samples. This is likely in response to environmental stress. Since such linkages are partially retained after the purification–3-day culture process (cf. Figure 9h), it is also suggested that such a process is not capable of returning the cells to their standard structure. This outcome is similar to what was observed in the analyses of oxidized sulfur compounds and cytochrome molecules (in Section 3.2 and Section 3.3, respectively). We will further discuss this point in Section 3.10.

Data published by Gantner et al. [68] showed that the β–1,3–glucan content of yeast and hyphal cells is similar, which strengthens our hypothesis that the observed increase in *R_G__β/α_* samples in the clinical samples is due to biofilm rather than to a switch in the morphogenic state of the cells. However, the present data also suggest that *Candida* cells exposed to the human body interactively use cell wall α–glucan structures as effective ploys in survivorship, tailoring water insolubility through enhancing the fraction of α–1,3–glycosidic linkages. The possibility that such changes are related to a yeast-to-hypha morphogenic switch will be discussed in a later section. Note, finally, that the differences in glucan architecture of both EMC and cell walls are mainly responsible for the morphological differences detected in Zone IV between the spectra recorded on as-collected and purified/3-day-cultured samples (cf. Figure 3i–p and Figure 4c,d,g,h).

### 3.5. Raman Characterization of Chitin Structures in Biofilm and Cell Walls

An additional consideration regarding the *Candida* cell wall structure is that α–1,3–glucans, namely the most rigid molecule among polysaccharides, intertwine with chitin (another quite-stiff molecule) to form a rigid and hydrophobic scaffold [69]. It is indeed well known that Candida cells, under an environmental trigger, could be quite efficient in minutely monitoring the integrity of the protective glucan–chitin structure of their cell walls and in tailoring it in accordance. Regulation of cell wall flexibility permits the maintenance of a delicate balance between rigidity and compliance, thus enabling the modulation of both turgor-driven and osmotic stress-driven cell expansion to prevent breakage [69]. With the above notions in mind, we monitored the characteristic Raman fingerprints from the chitin structure in the Raman spectra of clinical samples upon following a procedure previously optimized for reference Candida samples [7,8,9,10]. Two peculiarities of the chitin structure among other polysaccharides consist of the presence of C–O–C bonds between neighboring rings and the appearance of amide groups to partly replace OH groups as structural sub-units in the polymeric chain [70]. The distinctive vibrational characteristics that enable one to assess these structural peculiarities are (i) a doublet at 1057 and 1109 cm^−1^ (C–O–C stretching band within and between rings, respectively; Figure 10a) and (ii) a doublet at ~1644 and 1660 cm^−1^ (Amide I region), which has been associated with two different types of hydrogen bonds present in the α–chitin crystals, namely, the intermolecular (C=O^…^HN) and the intramolecular (C=O^…^HO(C6) or C=O^…^HN) ones (cf. schematic draft in Figure 10b) [71,72,73].

As suggested in previous work [7], the Raman band intensity ratio, *R_E/R_* = *I*_1109_/*I*_1057_, also referred to as the “esterification ratio”, could be taken as representative of the relative fractions of ether vs. ring C–O–C bonds in α–chitin. It follows that α–chitin structures with longer chains are expected to contain a higher fraction of ether C–O–C bonds and a higher crystallinity; the *R_E/R_* ratio can thus be assumed as a marker for chitin’s degree of crystallinity. The higher the ratio the longer the crystalline chains. On the other hand, the analysis of the 1644 and 1660 cm^−1^ doublet could be seen as a means to screen *Candida* samples according to their degree of acetylation. Note that the degree of acetylation, namely, the fraction of acetyl groups present in the α–chitin structure, is also linked to the crystalline structure of chitin. A high degree of acetylation favors a stable α–chitin crystalline structure through a network of hydrogen bonds that the amine acetyl groups can form with NH groups, while a low degree of acetylation is associated with a higher probability of primary amine groups in the chitin structure and, thus, favors more defective structures and/or is a marker for the presence of different chitin polymorphs (cf. inter- and intra-chain bond types in the α–chitin structure as reported in Figure 10b) [74,75,76]. As previously shown [7], the intensity ratio, *R_i/i_* = *I*_1644_/*I*_1660_, referred to as the Raman ratio of intermolecular-to-intramolecular hydrogen bonds, is a spectroscopic parameter quantitatively linked to the degree of acetylation of α–chitin in *Candida* cell structures. Data from reference α–chitin structures with known structural characteristics [72,77] showed that the higher the *R_i/i_* ratio, the higher the degree of chitin acetylation and, thus, the more ordered the α–chitin structure. We have previously validated the suitability of the *R_i/i_* ratio for quantitatively evaluating the chitin structures in standard *Candida* species with respect to the intrinsic degree of acetylation of their chitin structures in a stress-free state [7].

In this study, we attempted to extend to the clinical samples the above criteria of evaluation for α–chitin degrees of esterification and acetylation according to the Raman parameters, *R_E/R_* = *I*_1107_/*I*_1057_ and *R_i/i_* = *I*_1644_/*I*_1660_, respectively. Figure 10c–f show α–chitin sub-bands with, in the inset, the computed *R_E/R_* and *R_i/i_* ratios for standard *C. albicans* LSEM865, *C. glabrata* LSEM47, *C. parapsilosis* LSEM868, and *C. tropicalis* LSEM1823, respectively (i.e., as extracted from the deconvoluted spectra in Figure 5d,e, respectively). On the other hand, Figure 10g,h reports the same sub-bands and the respective *R_E/R_* and *R_i/i_* ratios (values in inset) for the clinical samples (cf. labels) as-collected from candidiasis patients and after purification–3-day culture, respectively. As seen, the standard samples all presented a similar esterification ratio, *R_E/R_* = 1.27~1.36, corresponding to degrees of crystallinity in the interval 22~25% [7]. However, as discussed in detail in a previous paper [7], *C. tropicalis* and *C. glabrata* possess ratios of intermolecular-to-intramolecular hydrogen bonds, *R_i/i_*, ranging between 0.60 and 0.67, which correspond to quite high degrees of acetylation (95~97%) compared to *C. albicans*, which is, instead, characterized by less-acetylated α–chitin structures (i.e., *R_i/i_* = 0.22, corresponding to a degree of acetylation ~78%). We newly report here that standard *C. parapsilosis* possesses an intermediate *R_i/i_* = 0.36, corresponding to a degree of acetylation of ~87%. A higher degree of acetylation for the same degree of crystallization might point to a non-negligible presence of chitin allomorphs different from α–chitin. Thus, this circumstance does not necessarily represent a more disordered state in the α–chitin allomorph itself. As a matter of fact, while intra-chain hydrogen bonds are present in all chitin allomorphs, the inter-chain ones are relatively rare in γ–chitin and completely absent in β–chitin [78,79,80,81], which is likely the reason why we see an apparently lower *R_i/i_* ratio in standard *C. albicans*.

Looking now at the chitin signals from clinical samples, one could immediately notice that the majority of samples in the as-collected state (Figure 10g) showed lower or similar *R_E/R_* ratios with respect to the respective standard samples. Exceptions to this decreasing trend were Samples 3 and 4 (assigned to *C. glabrata* and *C. albicans*, respectively), with higher *R_E/R_* ratios compared to the respective standard samples. Conversely, Samples 6 and 7 (assigned to *C. glabrata* and *C. albicans*, respectively), showed the lowest *R_E/R_* ratios among the as-received samples, at ~26% and 36% lower than those of the respective standard sample (corresponding to ~12% and ~18% decrease in crystallinity) [7]. After purification (cf. Figure 10h), the *R_E/R_* ratios were partly or fully restored to the values of the respective standard samples, again with the exception of Sample 6, which did not show any crystallinity increase (i.e., *R_E/R_* values equal to 0.94 and 0.93 before and after purification–3-day culture, respectively). The general trend for the *R_E/R_* ratio suggests that alterations of crystallinity are mainly related to the presence of ECM, which contains mixed biofilms, including *intra* but not *inter* C–O–C bonds. In support of this argument, one could note the distinctly high signals at 1057 and 1109 cm^−1^ in the biofilm sample isolated from *C. albicans* (cf. spectrum in Figure 9a).

Bacterial biofilms with different degrees of maturation, as produced by different pathogens (e.g., *Streptococcus mutans*, *Streptococcus gordonii*, *Streptococcus oralis*, *Porphyromonas gingivalis*, etc.), might also contribute to altering the α–chitin signals [82,83]. Staphylococcal biofilms generally contain poly-β–(1–6)–*N*–acetylglucosamine as a main exopolysaccharide component, also referred to as polysaccharide intercellular adhesin [84,85,86]. The Raman spectrum of the pure N-acetylglucosamine compound indeed presents *intra*-C–O–C bands very close to those of chitin [87]. It should not be a fortuitous circumstance that Sample 6, the only one showing a low but unchanged α–chitin crystallinity after biofilm purification, corresponded to the only completely toothless patient (cf. Table 1). Such a circumstance suggests a strong predominance of *Candida* cells over bacteria in the oral cavity with minimum interference of bacterial biofilm signals in the spectrum. Moreover, a complete predominance of *Candida* cells in the overall oral flora population might have triggered a peculiar cell wall structure by privileging flexibility above impermeability.

The general trend observed for the *R_i/i_* ratio revealed significantly increased values (up to five times) with respect to the respective standard samples (cf. Figure 10c–f), with even further enhancements upon purification–3-day culture (cf. Figure 10g,h). These trends cannot be easily explained only according to crystallization or chitin polymorphism. Neither could the overlapping effect of bacterial biofilm be invoked, since the exopolysaccharide poly-β–(1–6)–*N*–acetylglucosamine molecules retain a high degree of deacetylation during the formation of staphylococcal biofilms as a consequence of deacetylated form-mediating resistance to neutrophil phagocytosis [88]. A possible explanation for the apparently high degree of acetylation recorded in clinical samples (with the only exception being Sample 8) could be the presence of additional acetylated molecules or other molecules that enhance the intensity of the Raman signal at 1644 cm^−1^. This point will be further discussed in the next section.

In summary, while the *R_E/R_* ratio appeared to partly remain a valid parameter for estimating the balance between *inter-* and *intra*-C–O–C bonds, even if it includes bacterial biofilms, the presence and variability of biofilm structures (as a result of bacterial reaction to environmental factors) hampered a clear evaluation of the degree of chitin acetylation in the cell wall structure for the clinical *Candida* samples, even after sample purification. Finally, note that the above-described variations in α–chitin crystallinity are partly responsible for the negative spectral differences observed in Zone V between the collected and purified/cultured samples (cf. Figure 3i–p), while the anomalous trend in the Amide I spectral interval contributes to the observed positive difference in Zone VIII. Spectral differences in Figure 3 will be further discussed in the forthcoming sections, which consider the Raman markers of bacterial biofilms and the effect of morphogenetic switching on the spectrum of *Candida* samples.

### 3.6. Spectroscopic Fingerprints of Environmentally Driven Morphogenesis

Classified as a key virulence characteristic [89], morphogenetic switching from yeast to hyphal configuration generally has profound effects on the structure of a fungal cell’s surface. However, *Candida* species can generate subtly different forms of pseudohyphae and hyphae when embedded in different environments and react to a host immune response in a patient-specific manner. A number of different environmental inputs could activate hyphal development, such as low levels of nitrogen and related starvation [90,91,92], ambient pH [93,94], hypoxia and embedment in a matrix [95,96], and genotoxic insults acting through the pharmacological inhibition of DNA replication or DNA damage [97,98,99]. ROS generated by phagocytic cells induce genotoxic stress. Therefore, *C. albicans* hyphae escaping from macrophages are not structurally equivalent to hyphae that (unattacked by macrophages) invade mucosal surfaces, further complicating the host–pathogen interaction [99]. In other words, the yeast–hyphae population ratio is highly variable, and not all *Candida* hyphae are identical. In the above scenario, the morphogenic switch adds further variables to the build-up of the fungal cell surface, depending on the nature of the microenvironment. The Raman spectra recorded on clinical samples correspondingly reflect such complexity. With a number of concurrently possible contexts, the Raman spectroscopic challenge in the present analyses of clinical samples consisted of rationalizing the effect of morphogenesis on the Raman spectrum while locating spectroscopic parameters that could shed light on the complex array of environmental inputs.

Among the molecules that compose *Candida* cell walls, the boldest dependence on morphogenetic changes has been reported for α–chitin [100]. This polymorphic form of chitin was found to reach 3~5-times higher fractions in *C. albicans* hyphae than in the corresponding yeast clades [100,101,102]. The second in the rank of morphogenic variability are cell wall outer-layer mannans covalently associated with proteins to form glycoproteins. According to Machova et al. [103], yeast and hyphal cell walls of *C. albicans* present significant differences in the carbohydrate composition of mannoproteins. While yeast mannoproteins consist of up to 46% mannans, the mannan content only reaches ~14% in hyphal cells, with protein contents of 47~53% in yeasts vs. only 3~4.5% in hyphae. Moreover, mannans from yeast, independent of specific serotypes, were found to be more branched and contained higher amounts of mannose (>90%) than mannans from hyphae (66~76%). These characteristics suggest that the yeast-to-hypha morphogenic switch involves not only a significant *stiffening* of the cell wall structure but also a substantial *simplification* with respect to the mannan structure.

In order to substantiate and rationalize the above structural notions in terms of Raman parameters, we recorded Raman spectra from standard *C. albicans* LSEM865 at increasing steps of yeast-to-hypha morphogenic transformation (induced in vitro by genotoxic stress; cf. optical micrographs in Figure 11a–c, respectively). The spectra recorded for the pseudohypha (intermediate) and hypha (final) configurations are shown in Figure 11d,e, respectively. They should be compared with the reference (initial) yeast spectrum in Figure 5e. Spectral differences were computed upon subtracting the spectrum of pseudohyphae (in (d)) from that of hyphae (in (e)), as shown in the subtraction plot between Figure 11d,e. In this basic in vitro experiment, the spectral intensity recorded for pseudohyphae and hyphae samples in Zones IV and V showed a tendency to become increasingly stronger compared to that recorded for the yeast sample in the same zones. Chitin signals fall in these spectral zones. Therefore, their increase indeed links to a volumetric increase in chitin content upon morphogenic switching, which is in agreement with the literature [100,101,102]. Note also that the in vitro cultures in this basic experiment lacked the presence of any commensal bacterium. Therefore, the observed difference in Zones IV and V should be ascribed to the structural differences intrinsic to the cell walls.

In light of the basic experimental data in Figure 11, let us now consider the trend for clinical samples. The majority of them showed stronger signals in Zones IV and V after purification–culture compared to the respective as-collected samples (cf. subtraction plots in Figure 3i–p). Considering what we observed in the in vitro experiment of morphogenic switching in Figure 11, this trend does not support the interpretation that the adopted purification–3-day-culture procedure reduced the amount of α–chitin (e.g., its C–O–C signals in Zone V at 1057 and 1107 cm^−1^) and/or the fraction of hyphae, in contradiction with the microscopy evidence shown in Figure 4a,b,e,f. This discrepancy could only be explained by invoking a preponderance of overlapping C–C or ring signals from other biomolecules (e.g., lipids, phenylalanine, etc.) in these spectral zones. Looking into further details of the spectral differences between Zones IV and V, we examined the spectroscopic features of mannan chemistry included in these zones. The spectral region between 950 and 1200 cm^−1^ is characteristic of deformation modes of C–O–C, C–OH, C–CH, and O–CH groups and displays bands representative of ring conformation and related orientation [104]. Unlike standard *Candida* samples [8], the vibrational response of clinical samples in this wavenumber interval was clearly the result of complex signal overlap from a number of different molecules, which, again, might include lipid chains also contributing signals in other spectroscopic regions (i.e., 1400~1500 cm^−1^ and 1250~1300 cm^−1^) [105]. As seen in the plots in Figure 3i–p, subtracting the purified/cultured yeast spectra from the as-collected ones gives a positive difference in the correspondence of the spectral interval of950~1000 cm^−1^ in Zone IV (i.e., at ~960 and ~975 cm^−1^, where two C–O–C main bands for α– and β–mannopyranose rings are located) [8] for all samples, except for Sample 6. If purification–culture is expected to induce a switch back from hyphae to yeast in the clinical samples, the observed positive difference actually points to an inverse path with respect to any “*simplification*” of the mannan structure expected for the hyphal morphology. Note also that any “*simplification*” trend could not be observed in the in vitro experiment of the morphogenic switch in Figure 11.

In summary, because of complex overlaps of Raman signals from a variety of different molecules, Zones IV and V could hardly serve as spectroscopic marks to assess the volumetric increase and reduction in α–chitin and α/β–mannopyranose fractions in clinical samples. This circumstance, in turn, rules out the possibility of quantitatively determining the extent of hyphal transformation in clinical samples from Raman signals in those spectral zones.

### 3.7. Spectroscopic Fingerprints of Peptide Toxins

After failing to use cell wall structural signals to spectroscopically resolve yeast-to-hypha (or the inverse) switching and quantify the pathogenic impact of *Candida* samples retrieved from patients, we looked for the spectral fingerprints of candidalysin, the amphipathic peptide toxin secreted by *C. albicans* upon morphogenic switching into hyphae [106,107,108,109,110,111]. In candidiasis, *C. albicans* induces epithelial invagination with the formation of an invasion pocket surrounded by the host cell membrane, in which it secretes candidalysin peptide molecules. These molecules take a roundish architecture that produces pores and ultimately results in cell damage and death [111]. Figure 12a shows a schematic draft of the structure of candidalysin [110], while in (b), we recorded the Raman spectrum of the pure candidalysin peptide. To our knowledge, this is the first reported Raman spectrum for pure candidalysin without employing surface enhancement procedures that could alter its structure. We performed band deconvolution (cf. Figure 12b) upon introducing in the deconvolutive machine-learning algorithm the basic molecular components reported in the literature [110,112]. The strongest bands in the candidalysin spectrum were observed at 661, 987, 1438, and 1650 cm^−1^ (cf. Figure 12b), which correspond to C–S stretching peculiar to methionine, C–C and C–N stretching, CH_2_ scissoring, and Amide I in the α–helix, respectively. Upon comparing Figure 11d,e, in which spectra from *C. albicans* are reported with proceeding morphogenic switching into hyphae, significant enhancements could indeed be observed for bands at ~661, 987, and 1650 cm^−1^ (i.e., ~220%, 110%, and 46%, respectively), while a slight reduction was observed for the band at 1438 cm^−1^ (~5%). We hypothesize that the intensity increase of the three above bands is the result of the formation of candidalysin toxins upon hyphal switching, while the slight reduction of the 1438 cm^−1^ band could be ascribed to the concurrent volumetric reduction of other molecules containing CH_2_ units (e.g., according to the structural changes discussed in Section 3.5). Note also that the spectrum of candidalysin in Figure 12b revealed a significantly strong shoulder exactly at 1644 cm^−1^ (labeled with an asterisk). We hypothesize that such overlap is the main reason for an altered Raman intensity, corresponding to intermolecular (C=O^…^HN) bonds in the α–chitin crystals that apparently enhanced the *R_i/i_* acetylation ratio recorded in clinical samples.

Figure 12c,d shows the 661, 987, and 1650 cm^−1^ spectral components extracted from spectra of standard *C. albicans* LSEM865 before and after full morphogenic switching, while Figure 12e,f display the same bands for all clinical samples before and after culture–3-day purification, respectively (cf. labels in inset). A Raman spectroscopic parameter linked to a candidalysin increase in the samples could be conceived of as *R_Clys_* = (*I*_661_ + *I*_987_ + *I*_1650_)/(*I*_661_ + *I*_987_ + *I*_1650_)*_ref_*, where the subscript *ref* refers to the sub-band intensities in the spectrum before morphogenic switching (in Figure 12c). This spectroscopic parameter could be considered as an index for virulence. The higher the *R_Clys_* value, the higher the amount of candidalysin present in the sample and, thus, the virulence. The computed *R_Clys_* ratio for each clinical sample is given in the inset to each plot in Figure 12e,f. As seen, the majority of the as-collected *C. albicans* samples showed that the *R_Clys_* values comprised between 1.0 and 1.8 (cf. Figure 12e). Similarly, also, Sample 8, assigned to *C. tropicalis*, showed a higher value (*R_Clys_* = 1.4). On the other hand, no candidalysin virulence was found for the *C. parapsilosis* sample (i.e., Sample 5) and low or no virulence in the *C. glabrata* samples (i.e., Samples 3 and 6), for which the *R_Clys_* value was recorded as 1.1 and 1.0, respectively. The above results were in line with the published literature according to which the cytolytic peptide toxin candidalysin can be produced in a similar way by *C. albicans* and *C. tropicalis*, despite having different amino acid sequences and a higher damaging potential in the latter [113]. On the other hand, both *C. glabrata* and *C. parapsilosis* do not harbor candidalysin orthologs within their genomic sequences, thus implying that both the damage potential and pathogenicity in these species depend on different fungal factors [113]. Therefore, the present Raman data confirmed that *C. glabrata* and *C. parapsilosis* cannot produce candidalysin and, thus, do not cause mucosal disease because of its secretion. Upon purification–culture, the *R_Clys_* ratios of all virulent (as-collected) *C. albicans* samples returned to values close to one, thus showing that the purification process has devoided the samples of candidalysin, and the 3-day culture has, at least partly, re-established the yeast polymorph.

### 3.8. Statistical Validation by Means of Raman Imaging

Raman imaging was conducted on *Candida* samples in order to validate the Raman results collected on the average spectra. The Raman spectra so far discussed were obtained by averaging over tens of spectra collected at different locations with relatively low spatial resolution (20× optical lens) to cover a total area of the sample on the order of ~1 mm^2^ with high spectral resolution (better than ±1 cm^−1^). On the other hand, the Raman imaging approach enabled the collection of a much higher number of spectra (in the order of 10^6^) over a smaller area (~10^3^ μm^2^), with a high spatial resolution (spots of single μm^3^ with sub-micrometer displacements) but a lower spectral resolution (±2 cm^−1^). The imaging approach, in addition to being complementary to and statistically validating the spectral averaging approach, revealed details of spatial distribution for selected molecules. We searched for and imaged specific zones in the as-collected samples that, low in biofilm, predominantly consisted of *C. albicans* cells in yeast and had a hyphal morphology (Figure 13a,b), respectively). Spectral wavenumbers at 520 ± 20, 661 ± 2, 987 ± 2, and 1650 ± 2 cm^−1^ (in Figure 13c,d, Figure 13e,f, and Figure 13g,h,i,j, respectively) were monitored. As seen from the comparison, Raman imaging gives a vivid example of how morphogenic shift into the hyphal form affects the chemical composition of cells. Imaging the Raman intensity in the spectral interval 520 ± 20 cm^−1^, namely in the interval mainly contributed by disulfide bonds, revealed that yeast cells were subjected to quite high levels of physiological stress (from ROS and/or pH alteration) (cf. Figure 13c). Such a stress state appeared to be conspicuously released when the cells switched into hyphal conformation (cf. significantly lower Raman intensity in Figure 13d). Conversely, significantly stronger Raman intensities were recorded for cells in the hyphal configuration in all three spectral intervals, namely 661 ± 2, 987 ± 2, and 1650 ± 2 cm^−1^. As discussed in Section 3.6, these bands are markers for candidalysin molecules and, thus, for cells’ virulence (cf. Figure 13e,g,i and Figure 13f,h,j for the yeast and hyphal configurations, respectively). Note that signals from candidalysin could also be observed in areas relatively far from the hyphal tips, meaning that these molecules were spread over the entire volume sample. The results of Raman imaging could be interpreted as a confirmation of the reliability of the Raman method and its suitability to systematically analyze, at the molecular level, both the stress state and the degree of virulence in clinical samples of oral candidiasis.

### 3.9. Other Overlapping Signals Characteristic of Bacterial Biofilms

The main signal mismatches between Raman spectra before and after purification–3-day culture (as seen in Figure 3i–p) can be related to the zones predominantly contributed by lipids, amide, and carbohydrate signals from bacterial biofilms. The largest influence from lipids in the Raman spectrum of *Streptococci* was reported to arise from the presence of hydrocarbon chains [114]. These signals are mainly detected in the spectroscopic regions of 1050~1200 cm^−1^ and 1250~1300 cm^−1^ (Zone V) and 1400~1500 cm^−1^ (Zone VII) [105]. However, the elimination of biofilm involved a significant signal reduction for all samples only in the latter, namely Zone VII. The strongest signals in this spectral zone are seen at 1445~1461 cm^−1^ and can be assigned to saturated lipids [115]. The present data are in agreement with the Raman measurements on *C. parapsilosis* and *Staphylococcus epidermidis* biofilms published by Hrubanova et al. [116] Another signal that was conspicuously eliminated upon purification was the one at ~380 cm^−1^ (Zone I), which can be ascribed to the β–D–glucosides peculiar to *Streptococci* biofilms [115]. Note that the presence of biofilms from Gram-positive bacteria is also expected to enhance Raman signals at 880~980 cm^−1^ (Zone IV) related to rhamnose, galactose, and glucose [117]. This trend was indeed observed for Samples 2, 4, and 5 (cf. Figure 3j,k,l, respectively). The spectral range 790~950 cm^−1^ (i.e., across Zones III and IV) was also indicated as comprising Raman signals that arise from side-group deformations of carbohydrates peculiar to bacterial biofilms [118]. The difference between Raman signals before and after sample purification confirmed this assignment, except for Samples 3 and 6, as indeed expected for specimens from patients with no or almost no residual teeth. Since ECM is a mixture of biofilms from *Candida* cells and bacteria in variable fractions, signal profiles from sugar molecules, which are present in any biofilm, but with different contents of glucans and glucose, can hardly be rationalized according to a general trend. For this reason, one could observe both positive and negative spectral differences by subtracting the Raman spectra from as-collected and purified samples. Such an indetermination mainly impacts Zones III and IV (cf. Figure 3i–p) and explains why the analytical approach based on PCA cannot directly be used to characterize the clinical samples of candidiasis infection.

### 3.10. Raman Spectroscopic Criteria for Assessing Candidiasis Severity

The findings in this paper have revealed that the diversity and variability of clinical candidiasis samples can hardly be captured by Raman analyses of “standard” (reference) *Candida* strains, and therefore, such an approach cannot be used to assess the degree of isolate virulence. The oral microbiota, in turn, is also influenced by various host-specific factors, such as age, genetics, immune status, and overall health, and builds up unique microbial communities with *Candida* cells, whose composition could be extremely diversified. Accordingly, merely matching spectra from clinical isolates to those of standard reference strains hardly helps in understanding virulence or gives hints for potential disease implications and treatments.

Our previous studies of standard *Candida* clades/subclades [7,8,9,10] located a number of different Raman parameters, which enabled the assessment of the carbohydrate and lipid structures of cell walls, physiological sulfur chemistry, and overall stress state. The present study, which attempted to directly apply those parametric assessments to clinical isolates and to compare them to standard samples, conspicuously failed as a consequence of host-variability factors and the presence of preponderant Raman signals from bacterial biofilms (e.g., sulfur compounds, glucans, and other exopolysaccharides). However, three Raman spectroscopic parameters appeared to survive such sample complexity, namely, *R_G_*_α3/α6_, *R_Cyt_*, and *R_Clys_*, as representative of cell wall rigidity/impermeability, cell physiological stress state, and morphogenic switch/virulence, respectively. These three parameters survived because the Raman signals involved in their computation are peculiar to *Candida* cells (i.e., not found in bacteria or biofilms) and/or of relatively high intensity in the overall sample spectrum.

Figure 14 shows the reciprocal dependences of the *R_G_*_α3/α6_, *R_Cyt_*, and *R_Clys_* parameters, which hint at peculiar aspects of sample diagnostics. In (a), a plot is given of the virulence factor, *R_Clys_*, as a function of the physiological stress factor, *R_Cyt_*, while (b) gives the cell wall structural parameter, *R_G_*_α3/α6_, as a function of the same abscissa *R_Cyt_*. Both plots include data from clinical samples before and after the purification–culture procedure in comparison with data from the reference *Candida* clades (cf. legends in inset). The *R_Clys_* vs. *R_Cyt_* plot (in Figure 14a) suggests the existence of an interval of the physiological stress state with a threshold value (*R_Cyt_*~0.75) above which the *Candida* cells sharply increase their virulence upon switching into hyphae and producing candidalysin. The majority of the clinical isolates trespassed such a threshold, while all samples after purification–culture turned back to a state of physiological stress. On the other hand, the *R_G_*_α3/α6_ vs. *R_Cyt_* plot (in Figure 14b) follows a linear trend over the entire interval of stress enhancement. This behavior can be interpreted as a passive reaction of *Candida* cells to the environmental burden, which is limited to self-protection.

Overall, the data in Figure 14a,b hint at a trend of “fight or flight”, in which the *Candida* cells initially cope with the environmentally driven stress state by building up increasingly stiffer and impermeable walls (i.e., the “flight” stage) and, then, beyond a certain stress threshold, react by morphogenically switching into hyphae and produce peptide toxins (i.e., the “fight” state). Note, finally, that the present data appear to support and complete the findings of a recent study published by Zeng et al. [119], in which the critical role of cytochrome *c* (here represented by the Raman parameter *R_Cyt_*) in regulating morphogenesis and in maintaining the virulence of *C. albicans* was elucidated by means of transposon-mediated genome-wide mutagenesis and transcriptomic analyses.

In summary, the three Raman parameters, *R_G_*_α3/α6_, *R_Cyt_*, and *R_Clys_*, once precisely evaluated with sufficient statistics in a given clinical sample, could stigmatize the extent to which the *Candida* cells contained in the sample are edging toward a threshold for morphogenic switching, maximum cell wall stiffness, and full virulence.

### 3.11. Limitations of This Study and Directions for Future Work

The main limitation of this study resides in the limited number (eight) of clinical samples investigated. The Raman algorithms proposed should, thus, be statistically validated on a large number of clinical samples in future work. Additional directions for future works could be in (i) comparing the values experienced by the three Raman parameters, *R_G_*_α3/α6_, *R_Cyt_*, and *R_Clys_*, on the same sample before and after topical or systemic antifungal medicines and biocides; (ii) comparing the parameters for *C. albicans* from healthy subjects with no oral fungal disease; and (iii) comparing local microenvironments, e.g., yeasts/fungi from dentures/appliances versus those from the tongue or other oral mucosal sites of infection.

## 4. Materials and Methods

### 4.1. Clinical Samples and Assessment of Their Clinical Characteristics

Eight samples for Raman analyses were obtained by swabbing the tongue mucosa of patients with oral candidiasis (3 men and 5 women aged between 39 and 92) with a sterilized cotton swab and then smearing them onto a glass substrate. A list of the investigated clinical samples and their respective clinical characteristics according to standard examination procedures (as explained later) is given in Table 1.

Portions of the clinical samples were then smeared on a glass substrate (MatTek Corporation, Ashland, MA, USA) and observed with a laser microscope (VK-x200 series, Keyence Co., Ltd., Osaka, Japan) at a magnification of 400~1000×. Cells from the reference *Candida* species were observed with differential interference contrast (DIC) microscopy (BX53, OLYMPUS Co., Tokyo, Japan) at magnifications of 400~1000×.

The clinical samples were then cultured in BD BBL™ CHROMagar™ (Franklin Lakes, NJ, USA) for 3~5 days to identify species and further cultured in Candida selective medium Sabouraud Agar Candida detector (Kamemizu Chemical Ind. Co., Ltd., Neyagawa, Osaka, Japan) for 3 days at room temperature under atmospheric pressure in order to measure the number of colonies. The selectively cultured samples were also swabbed and smeared onto a glass substrate to produce Raman samples, then fixed in paraformaldehyde/PBS (Nacalai Tesque, Inc., Kyoto, Japan) and washed in a phosphate-buffered saline (PBS) solution (Nacalai Tesque, Inc., Kyoto, Japan).

The collected samples were smeared on the respective culture media, streaked, and then incubated at 35 °C for 24~72 h (aerobic bacteria—carbon dioxide culture in chocolate agar with selective medium, anaerobic bacteria—anaerobic culture in Anaero Columbia Agar with Rabbit Blood, and Candida—aerobic culture in Sabouraud agar). The aerobic bacteria and anaerobic bacteria were primarily identified using a mass spectrometer, MALDI Biotyper ^®^ (Bruker, Billerica, MA, USA).

It should be noted at the outset that, in healthy subjects, oral biofilms mostly consist of oral bacteria with smaller fractions of fungi and human cells (human-derived epithelium). However, in patients affected by candidiasis, the fraction of fungi was generally quite preponderant, although the presence of bacteria cannot be neglected. In particular, *C. albicans* is known to form abundant biofilm in synergy with *Streptococcus mutans* (all samples examined in this study indeed included one or more populations of *Streptococci*; cf. Table 1) [120].

In a further attempt to isolate only the *Candida* pathogens, we employed a fungal selection medium on the swab samples in order to isolate them and to increase only the fungal population.

### 4.2. Reference Samples

Reference samples of *Candida albicans* LSEM865, *Candida glabrata* LSEM47, *Candida parapsilosis* LSEM868, and *Candida tropicalis* LSEM1823 were obtained from Teikyo University Institute of Medical Mycology (Hachioji, Tokyo, Japan). The Faculty of Nutrition, Kagawa Nutrition University in Saitama, Japan, provided a standard sample of *C. albicans*-derived β–glucan. *C. albicans*-derived β–glucans are known to stimulate peripheral pain-sensing nerves by promoting the release of ATP-secreting granules, relieving pain and itching [121]. Upon analyzing and comparing the molecular structures of commercially available β–glucans (i.e., from black yeast) and *C. albicans*-derived β–glucan using Raman spectroscopy, we searched for pathogenic molecules specifically expressed by *C. albicans* in its derived β–glucan. Additionally, it was first reported in 2016 that *C. albicans* produces the peptide toxin candidalysin [106]. Candidalysin damages epithelial cells and promotes inflammation, being also involved in the exacerbation of inflammatory bowel disease (IBD) patients [107,108,109,110,122,123]. We attempted here to provide a criterion to evaluate oral candidiasis from the relative expression intensity of the candidalysin contained in biofilms using Raman analysis. For comparison, the molecular structure of a commercially available standard sample of candidalysin (Peptide Institute, Inc., Ibaraki, Osaka, Japan) was also analyzed.

### 4.3. Raman Spectroscopy

Confocal Raman experiments were conducted by means of a Raman device (LabRAM HR800, Horiba/Jobin-Yvon, Kyoto, Japan) equipped with a holographic notch filter for high-efficiency/high-resolution spectral acquisitions. The wavelength of the incoming laser was 785 or 532 nm, with the laser source operating with a laser power of 70 and 30 mW, respectively. The Raman scattered light was analyzed with a double monochromator connected with an air-cooled charge-coupled device (CCD) detector (DV420-OE322, Oxford Instruments Andor Ltd., Belfast, Northern Ireland; 1024 × 256 pixels; grating resolution of 1800 gr/mm). The spectral resolution was ±1 cm^−1^. The acquisition time of a single spectrum was typically 20 s for 3 consecutive acquisitions at the same location to eliminate noise. The laser spot was ~2 µm, as focused on the sample through a 50× optical lens. Sets of 10 spectra were collected at different locations on each sample (over total areas of ~2 mm^2^) and averaged in order to obtain representative reference spectra. Two independent samples of each isolate were used to generate an average Raman spectrum. Raman spectra were subjected to baseline subtraction according to the asymmetric least square method. The average spectra were deconvoluted into a series of Lorentzian–Gaussian sub-bands using commercial software (LabSpec 4.02, Horiba/Jobin-Yvon, Kyoto, Japan). In performing this deconvolutive procedure, a machine-learning approach was applied, which employed an in-house-built automatic solver described in previous studies [7,15,124].

Raman imaging experiments were carried out with a dedicated Raman device (RAMANtouch, Nanophoton Co., Minoo, Osaka, Japan) operating in microscopic measurement mode (50× optical lens; numerical aperture, NA = 0.9) with high-precision confocal capability in two dimensions. A high in-plane spatial resolution of 300 nm could be achieved by exploiting a specially designed spectrograph with completely compensated aberration. Specially designed confocal optics also allowed for high spatial resolution (~670 nm) along the out-of-plane *z*-direction. This Raman microscope was also capable of ultra-fast simultaneous acquisition of up to 400 spectra, greatly reducing the laser irradiation time and enabling compatibility of the Raman scanning with the cells’ life. Selected excitation sources were 532 and 785 nm solid-state lasers operating with power at the sample surface of 10 and 190 mW, respectively. A 300 grating was used, which led to a spectral resolution of ±2 cm^−1^ (spectral pixel resolution of 0.3 cm^−1^/pixel), with an accuracy in laser spot spatial location of 100 nm. Raman hyperspectral images were generated by means of commercially available software (Raman Viewer, Version 1.4, Nanophoton Co., Minoo, Osaka, Japan). Lateral displacements involved steps of 500 nm for the laser focal point on the samples.

### 4.4. Statistical Analyses

Statistical analyses of Raman data sets were performed according to principal component analysis (PCA) on sets of 10 spectra for each clinical sample collected from patients and after purification–3-day culture. PCA analyses were conducted by means of the Origin software platform (Version 2022, OriginLab^®^ Co., Northampton, MA, USA) and were displayed as a set of “summary indices”, referred to as principal components.

## 5. Conclusions

A collection of eight clinical isolates from patients of oral candidiasis from two different *Candida* species was analyzed by Raman spectroscopy before and after a cycle of purification–3-day culture. The data were compared with reference *Candida* strains obtained from culture collections. Clinical isolates were observed by optical microscopies and verified to the species level by conventional identification methods. Two independent samples of each isolate were used to generate Raman spectra.

A high diversity and variability of oral candidiasis samples were found as a consequence of a variety of host-specific factors, which produced oral microbiota and ECM structures peculiar to each patient. This circumstance made useless any attempt to compare Raman multiomic analyses of clinical isolates to Raman analyses of the reference *Candida* strains. A careful look into the glucan structure of the clinical samples revealed that *Candida* cells underwent a complex defensive response to stress, which included an increasing synthesis of cell wall α–1,3–glycosidic linkages in a stress-dependent manner. The amount of this highly impermeable glucan structure accumulated in response to oxidative stress and could be quantified by means of the *R_G_*_α3/α6_ Raman parameter, which directly measured the relative increase in C–O–C glycosidic linkage stretching signals peculiar to the α–1,3–glucan chain. We also succeeded in a quantification of the level of cell physiological stress by monitoring the relative intensity of Raman signals from reduced vs. oxidized cytochrome *c* molecules and established a spectroscopic parameter, *R_Cyt_*, for its quantitative evaluation. One additional spectroscopic parameter, *R_Clys_*, could also be identified upon monitoring the Raman fingerprint signals of the peptide toxin candidalysin, which is only produced upon hyphal morphogenesis and, thus, directly linked to the yeast-to-hypha morphogenic switch. Upon relating to each other, for the trends observed for the three spectroscopic parameters, *R_G_*_α3/α6_, *R_Cyt_*, and *R_Clys_*, a “fight or flight” response was revealed to be in common with all *Candida* cell populations. Below a certain physiological stress threshold, yeast cells “flight” from antagonistic attacks upon abiding a merely passive reaction of cell wall reinforcement. However, beyond such a threshold, they switch into hyphae and “fight” back the antagonistic surroundings with eluting toxic molecules.

## Figures and Tables

**Figure 1 ijms-25-11410-f001:**
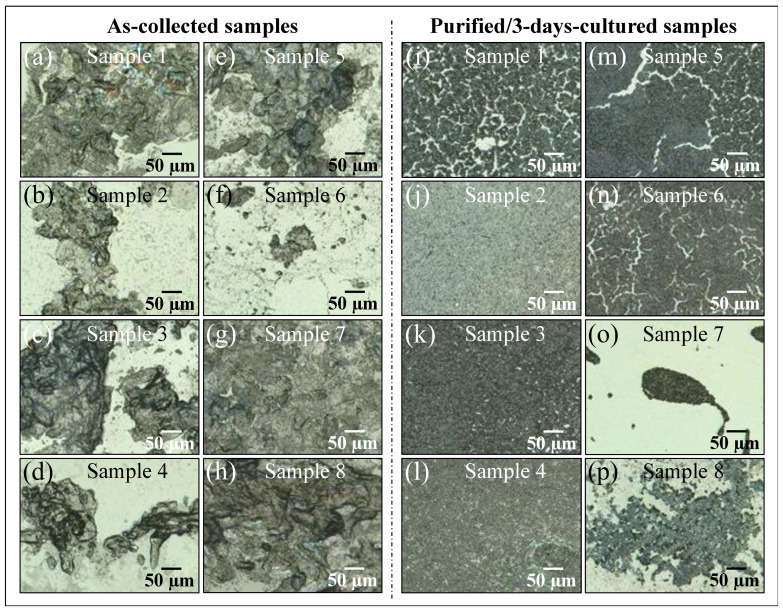
Laser micrographs of oral candidiasis samples: (**a**–**h**) clinical samples as collected from patients and (**i**–**p**) the same samples after purification and 3-day culture. Abundant presence of biofilm is observed in all as-collected samples, which is partially retained after purification and 3-day culture. Clinical details for all samples are given in Table 1.

**Figure 2 ijms-25-11410-f002:**
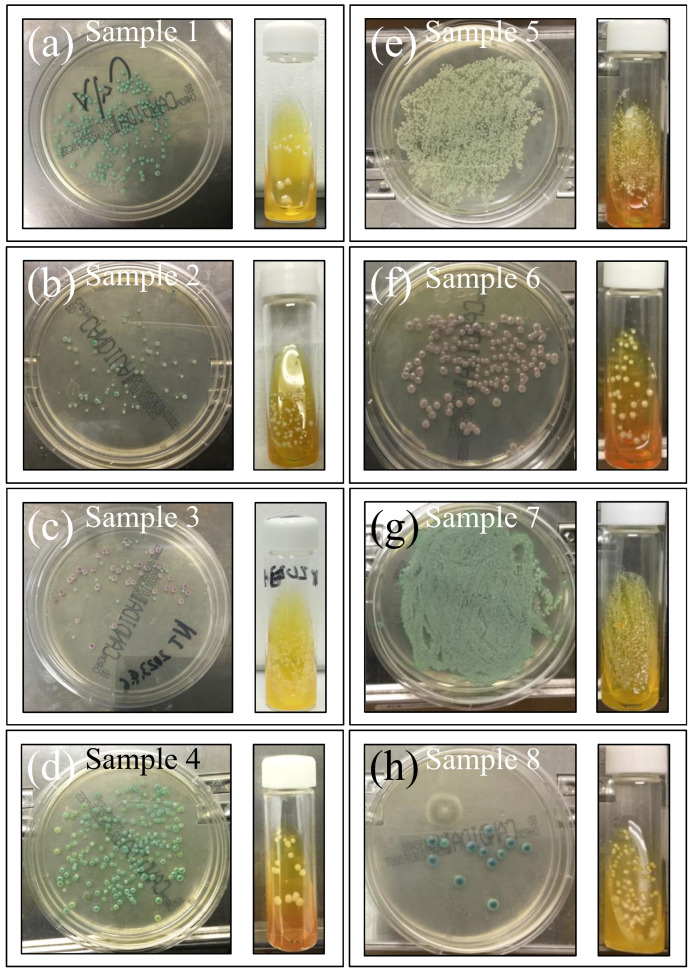
Swab samples cultured in Candida selective medium Candida detector (3 days) to measure the number of colonies and in BD BBL™ CHROMagar™ (for 3 days) to identify species. From the outputs of sample coloration in conjunction with microscopy observations, Samples 3 and 6 were identified as *C. glabrata* (cf. pink color), the former with a minor fraction of *C. albicans*. Sample 5 consisted of *C. parapsilosis*, while Sample 8 was mainly *C. tropicalis* with a minor fraction of *C. albicans*. All remaining samples belonged to the *C. albicans* species (cf. green color). Clinical details for all samples are given in Table 1.

**Figure 3 ijms-25-11410-f003:**
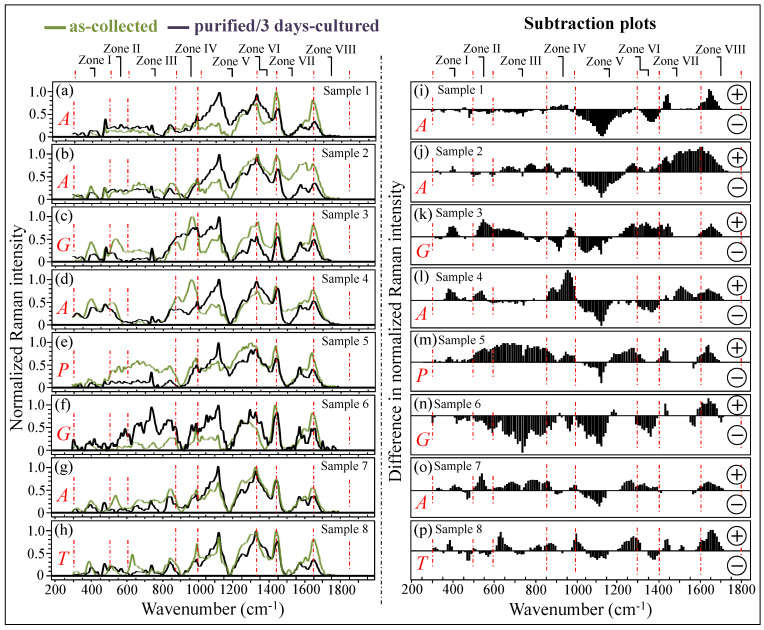
In (**a**–**h**), average Raman spectra for as-collected and purified/3-day-cultured samples in the wavenumber interval 300~1800 cm^−1^; the investigated interval was divided into 8 spectral zones at increasing wavenumbers (cf. labels in inset). Labels *A*, *G*, *P*, and *T* in inset refer to *C. albicans*, *C. glabrata*, *C. parapsilosis*, and *C. tropicalis*, respectively. The plots shown in (**i**–**p**) were obtained by subtracting the spectra of purified/3-day-cultured samples from those of the respective as-collected samples. The encircled signs + and − in inset refer to positive and negative spectral differences, respectively. Each spectrum was normalized to the maximum detected in the shown wavenumber interval. Red dotted lines delimit the eight spectral zones discussed in this paper.

**Figure 4 ijms-25-11410-f004:**
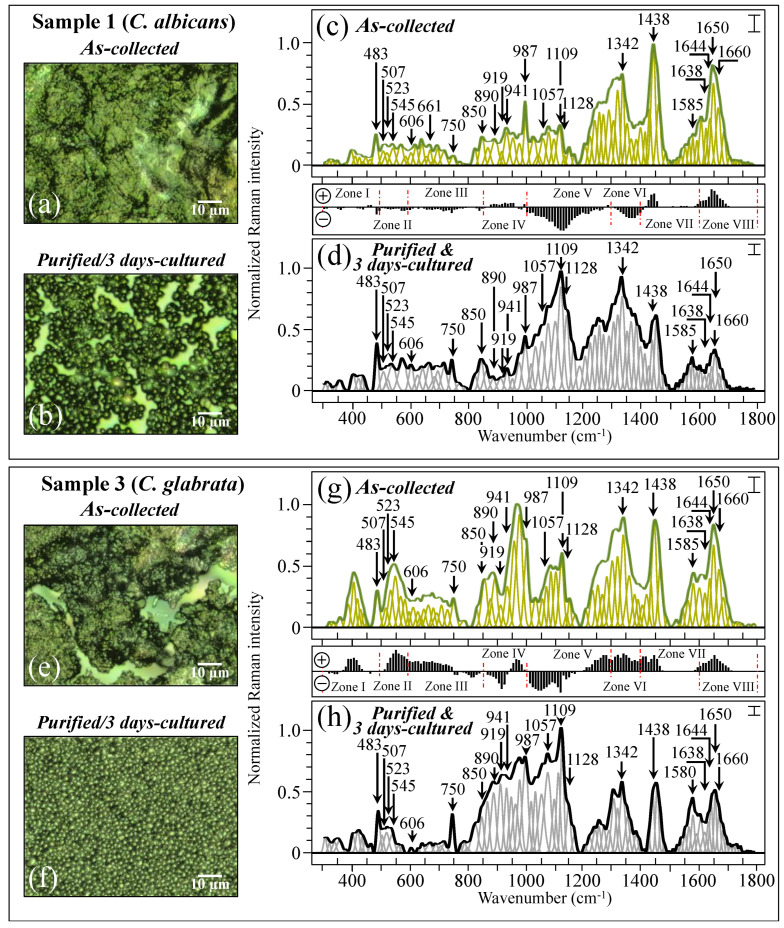
High-resolution laser micrographs of Sample 1 (*C. albicans*) as collected (**a**) and after purification–3-day culture (**b**). In (**c**,**d**), the respective average spectra (the same as shown in Figure 3a) are given after deconvolution into Gaussian–Lorentzian sub-bands (sub-bands in light green and grey colors, respectively). The subtraction plot between the two spectra is the same as that shown in Figure 3i. Similar characterizations are given for Sample 3 (*C. glabrata*): high-resolution laser micrographs in (**e**,**f**), average and deconvoluted spectra in (**g**,**h**) (the same as shown in Figure 3c), and the corresponding subtraction plot (the same as shown in Figure 3k) between the two spectra. The wavenumbers indicated in the inset are in cm^−1^ units (±3 cm^−1^) and refer to key spectroscopic signals used in Raman analyses. Each spectrum was normalized to the maximum detected in the shown wavenumber interval. The error bar in inset of each spectrum represents the standard deviation recorded over 10 spectra and recorded at different locations on the same sample. Red dotted lines are the same as those shown in Figure 3.

**Figure 5 ijms-25-11410-f005:**
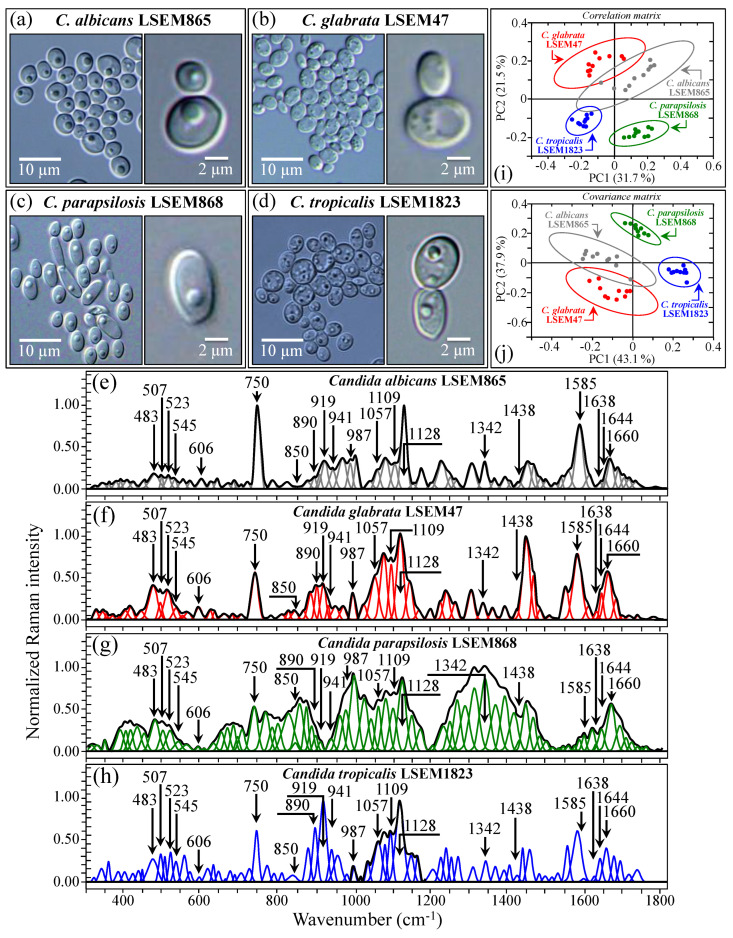
DIC micrographs of *C. albicans* LSEM865 (**a**), *C. glabrata* LSEM47 (**b**), *C. parapsilosis* LSEM868 (**c**), and *C. tropicalis* LSEM1823 (**d**) as taken from the respective reference samples (higher magnification DIC images in the respective insets). In (**e**–**h**), average (deconvoluted) Raman spectra are given for the above reference samples (cf. labels in inset; (**e**–**h**) are replotted from Ref. [7]). The wavenumbers indicated in inset are in cm^−1^ units (±3 cm^−1^) and refer to key spectroscopic signals used in Raman analyses. Plots in (**i**,**j**) represent three different combinations of PCA loading vectors, which refer to the entire spectral region 300~1800 cm^−1^ and allow for distinguishing between the above two reference *Candida* samples upon analyzing the mere morphology of their Raman spectra.

**Figure 6 ijms-25-11410-f006:**
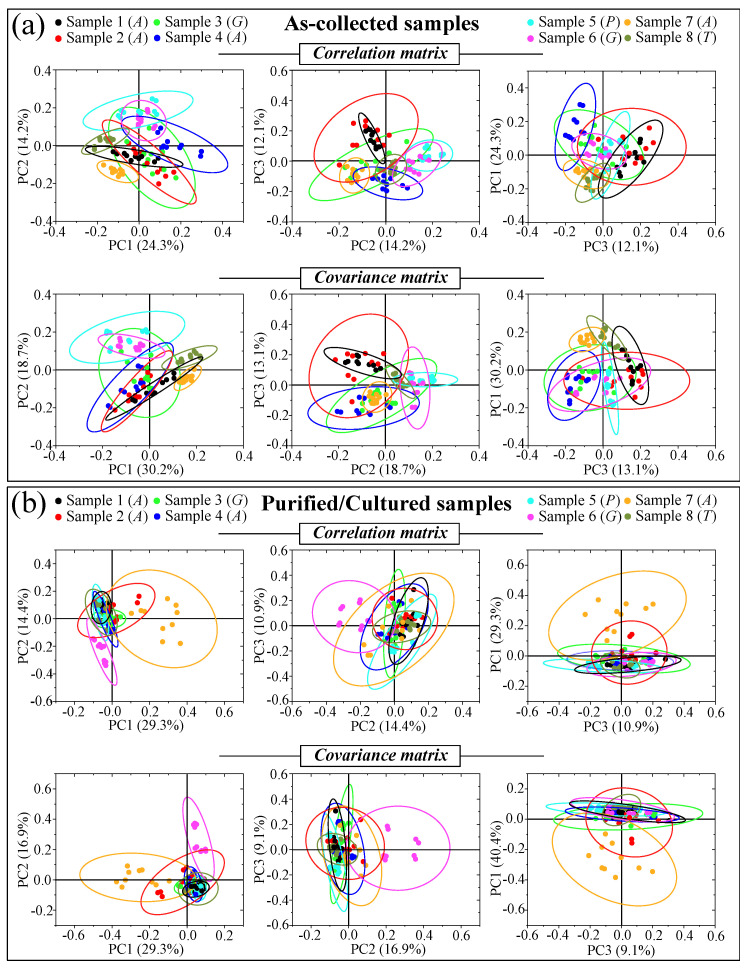
Plots for all as-collected (**a**) and purified/cultured (**b**) clinical samples are given for three different combinations of PCA loading vectors with respect to both correlation and covariance matrices (cf. labels in inset). All plots refer to the entire spectral region 300~1800 cm^−1^. Unlike the case of reference strains (cf. Figure 5i,j), none of the PCA plots, either for as collected or purified/cultured samples, allowed for distinguishing among different *Candida* species from the mere Raman spectral morphology.

**Figure 7 ijms-25-11410-f007:**
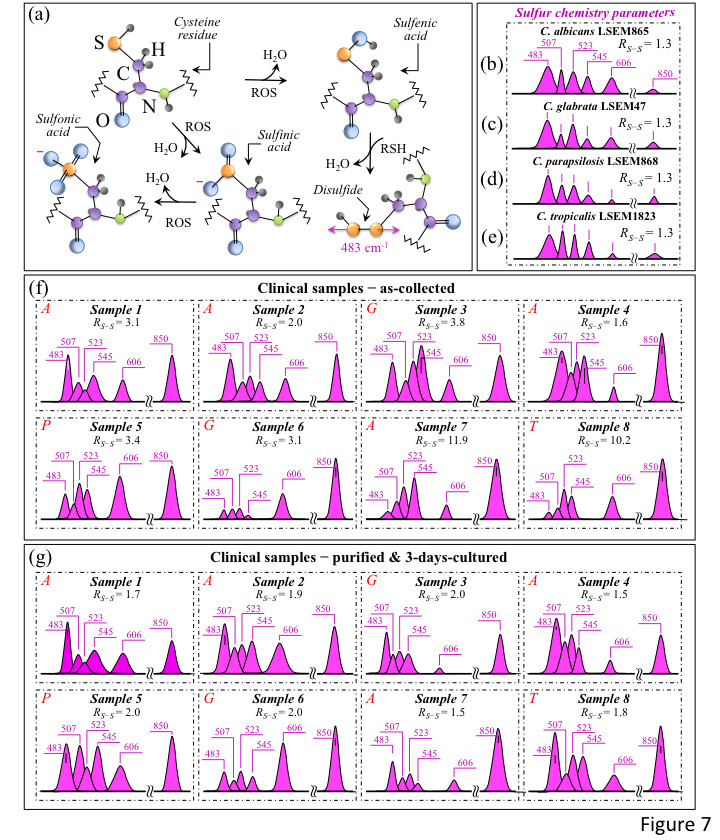
(**a**) Schematic draft for the suggested mechanism of cysteine degradation by ROS, which ultimately leads to hyperoxidation of sulfenic acid to irreversibly form sulfinic and sulfonic acids [38]. Spectroscopic difference in S–S bond structures observed upon extracting selected Raman bands (cf. text) from the spectra (in Figure 5) of *C. albicans* LSEM865 (**b**), *C. glabrata* LSEM47 (**c**), *C. parapsilosis* LSEM868 (**d**), and *C. tropicalis* LSEM1823 (**e**) reference samples are presented. The same procedure is applied to clinical samples in the as-collected (**f**) and purified/cultured (**g**) states, respectively (from spectra given in Figure 3a–h; cf. labels in inset). The spectroscopic ratio, *R_S-S_* = (*I*_507_ + *I*_523_ + *I*_545_)/*I*_483_, which was computed from the areas subtended by the selected Raman signals, gives the sulfide-to-sulfhydryl bond ratio (cf. values in inset). This parameter should, in principle, locate any eventual increase in the population of disulfide bonds in stressed *Candida* cells (cf. values in inset). However, the overlapping sulfur chemistry of bacterial population and related biofilm structures hamper this approach in the characterization of cells’ stress state. The wavenumbers indicated in inset are in cm^−1^ units and refer to key spectroscopic signals used in Raman analyses.

**Figure 8 ijms-25-11410-f008:**
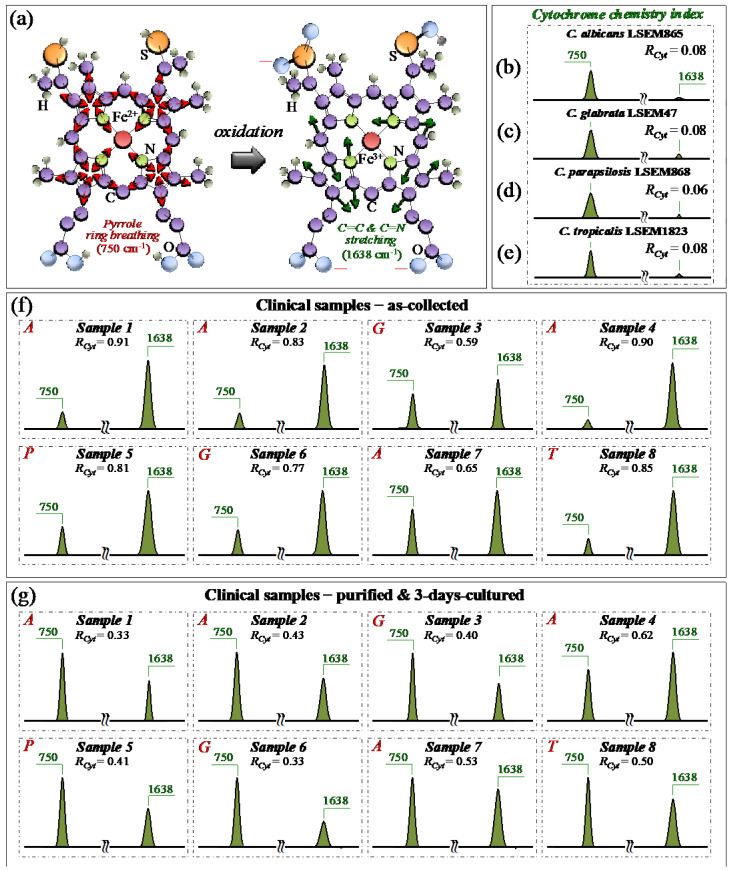
(**a**) Schematic draft for the mechanism of heme oxidation in cytochrome *c*, as a consequence of stress accumulation in *Candida* cells [46]. Spectroscopic differences for selected Raman signals from the heme molecule in the reduced and oxidized state were observed by extracting Raman bands at 750 and 1638 cm^−1^ from the spectra (in Figure 5) of *C. albicans* LSEM865 (**b**), *C. glabrata* LSEM47 (**c**), *C. parapsilosis* LSEM868 (**d**), and *C. tropicalis* LSEM1823 (**e**) reference samples. The same procedure was applied to clinical samples in the as-collected (**f**) and purified/cultured (**g**) states, respectively (i.e., from the spectra given in Figure 3a–h; cf. labels in inset). We computed the Raman spectroscopic ratio, *R_Cyt_* = *I*_1638_/(*I*_750_ + *I*_1638_), from the areas subtended by the selected Raman signals, which is representative of the level of physiological stress in the cells (cf. computed values in inset) [49,50,51,52]. The wavenumbers indicated in inset are in cm^−1^ units and refer to key spectroscopic signals used in Raman analyses.

**Figure 9 ijms-25-11410-f009:**
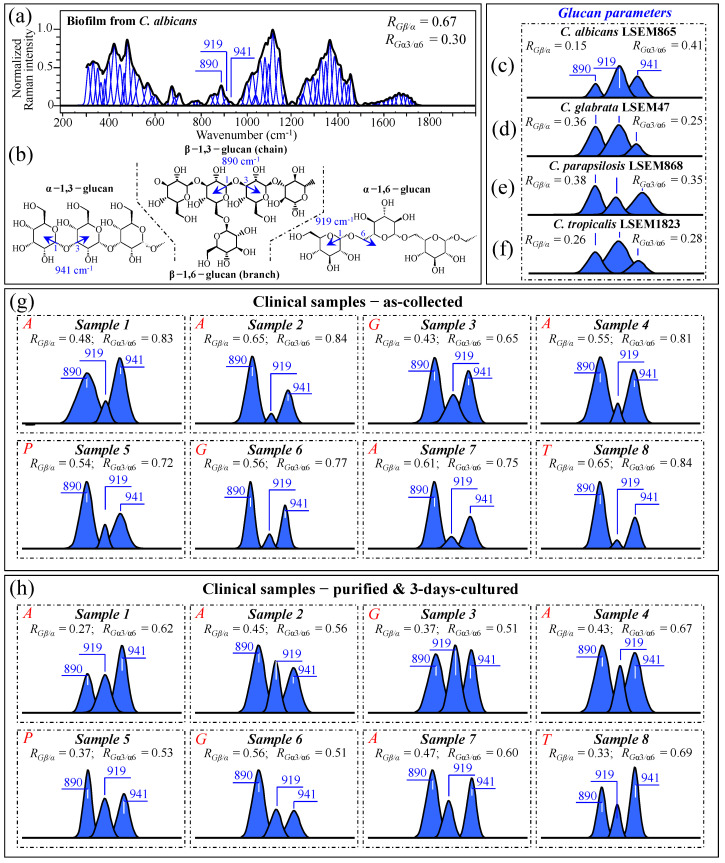
(**a**) Raman spectrum of the biofilm obtained from culturing *C. albicans* LSEM865 reference sample and (**b**) schematic drafts for β–1,3-linked chains (with β–1,6-linked branches), α–1,6–glucans, and α–1,3–glucans with their respective C–O–C glycosidic linkage stretching vibrations found at 890, 919, and ~941 cm^−1^, respectively (cf. labels in inset). Differences for selected spectroscopic signals were observed by extracting the above three C–O–C Raman bands from the spectra (in Figure 5) of *C. albicans* LSEM865 (**c**), *C. glabrata* LSEM47 (**d**), *C. parapsilosis* LSEM868 (**e**), and *C. tropicalis* LSEM1823 (**f**) reference samples. The same procedure was applied to clinical samples in the as-collected (**g**) and purified/cultured (**h**) states, respectively (i.e., from the spectra given in Figure 3a–h; cf. labels in inset). The Raman spectroscopic ratios, *R_G__β/α_* = *I*_890_/(*I*_890_ + *I*_919_ + *I*_941_) and *R_G_*_α3/α6_ = *I*_941_/(*I*_919_ + *I*_941_), were computed from the areas subtended by the selected Raman signals, which located the cell wall glucan composition in terms of α– vs. *β*–glucans and α–1,3- vs. α–1,6-linked glucans ratios, respectively (cf. computed values in inset). The wavenumbers indicated in inset are in cm^−1^ units and refer to key spectroscopic signals used in Raman analyses.

**Figure 10 ijms-25-11410-f010:**
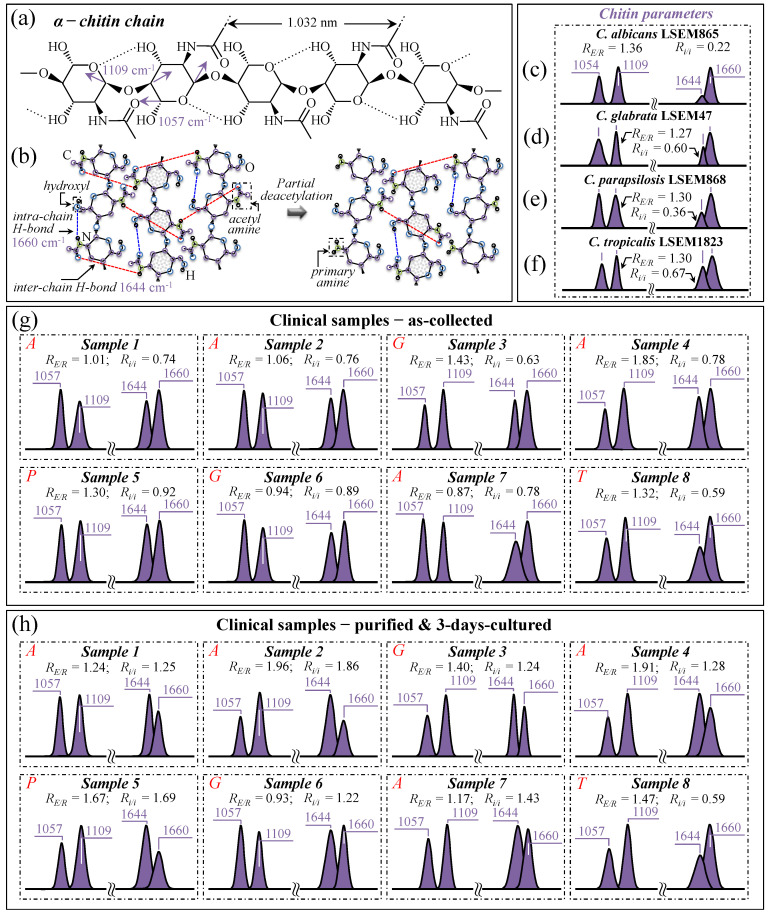
Schematic drafts of (**a**) the molecular structure of α–chitin chain and (**b**) the tertiary structure of α–chitin undergoing partial deacetylation; fingerprint vibrations and related wavenumbers are given in inset to both drafts. Spectroscopic differences in the selected Raman bands (cf. text) from the spectra (in Figure 5) are given for *C. albicans* LSEM865 (**c**), *C. glabrata* LSEM47 (**d**), *C. parapsilosis* LSEM868 (**e**), and *C. tropicalis* LSEM1823 (**f**) reference samples. The same fingerprint bands are retrieved from the deconvolution of spectra from clinical samples in the collected (**g**) and purified/cultured (**h**) states, respectively (i.e., from the spectra given in Figure 3a–h; cf. labels in inset). The spectroscopic ratios, *R_E/R_* = *I*_1109_/*I*_1057_ and *R_i/i_* = *I*_1644_/*I*_1660_, referred to as esterification and deacetylation ratios, respectively, were computed from the areas subtended by the selected Raman signals (cf. values in inset). These parameters should, in principle, locate eventual increases in cell wall rigidity in *Candida* cells from clinical samples compared to reference samples. However, the carbohydrate structure of bacterial biofilms overlaps the signals and hampers this spectroscopic approach in the characterization of cell wall structures (cf. text). The wavenumbers indicated in inset are in cm^−1^ units and refer to key spectroscopic signals used in Raman analyses.

**Figure 11 ijms-25-11410-f011:**
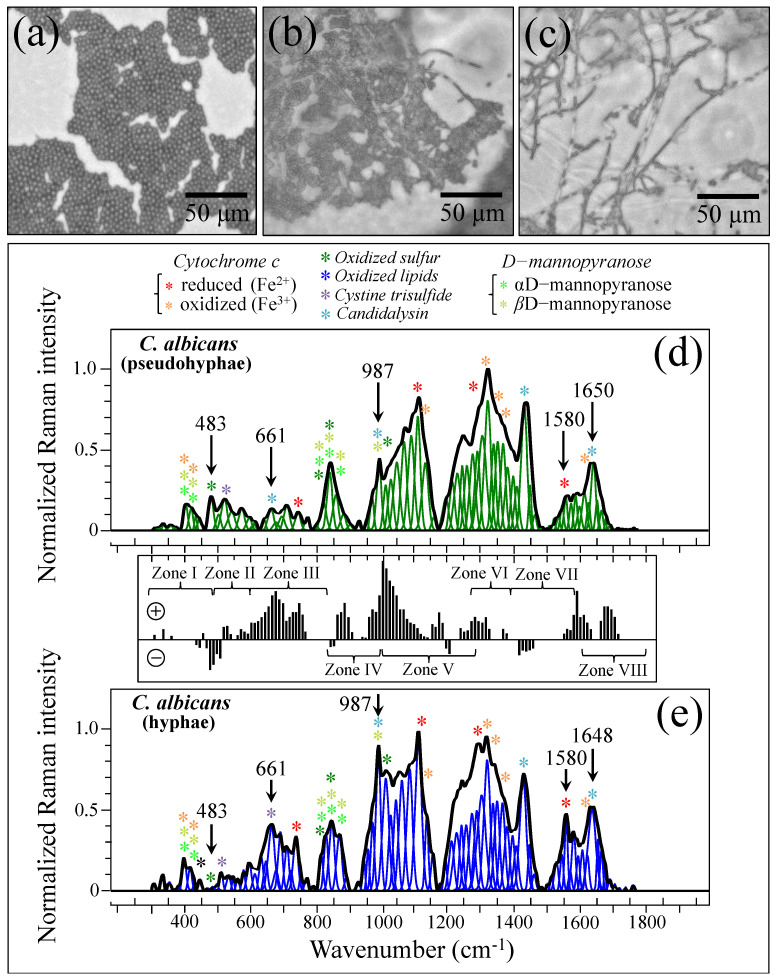
Optical micrographs of standard *C. albicans* LSEM865 at increasing steps of yeast-to-hypha morphogenic transformation: yeasts, pseudohyphae, and hyphae in (**a**), (**b**), and (**c**), respectively. Raman spectra recorded for pseudohyphal (intermediate) and hyphal (final) configurations are shown in (**d**) and (**e**), respectively. These spectra should be compared with the reference yeast spectrum in Figure 5c (spectral assignments as labeled in inset are from Refs. [7,8,9,10,34,35,36,37,38,39,52,53,54,55]). A plot of spectral differences, which were computed upon subtracting the spectrum of pseudohyphae in (**d**) from that of hyphae in (**e**), is given in the subtraction plot between (**d**,**e**) (spectral zones and symbols are the same as those given in Figure 3). The wavenumbers indicated in inset are in cm^−1^ units and refer to key spectroscopic signals used in Raman analyses.

**Figure 12 ijms-25-11410-f012:**
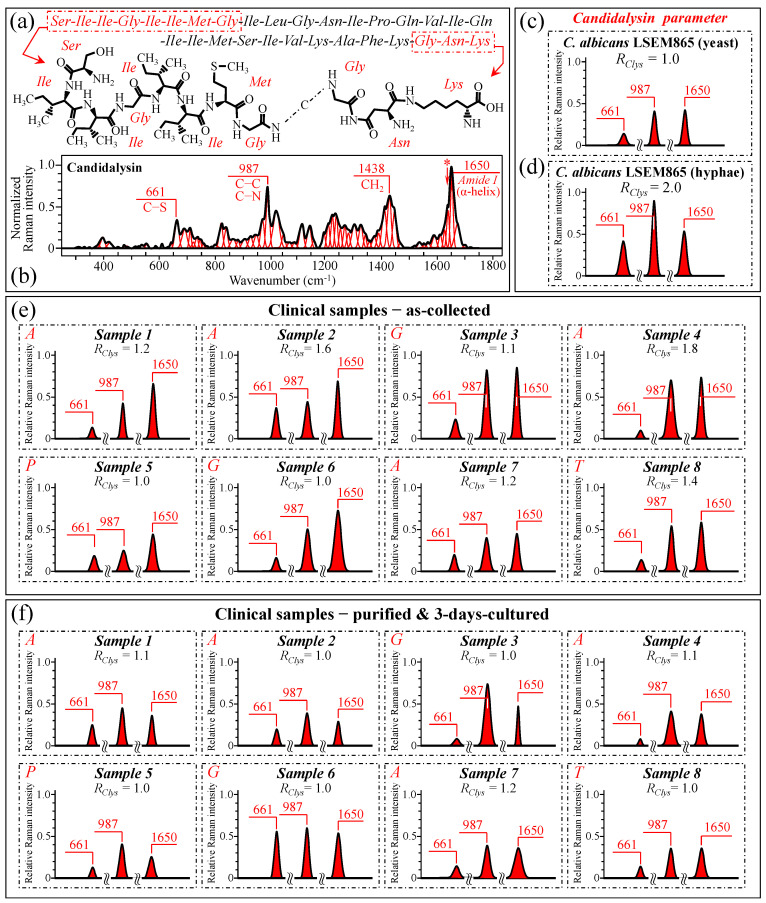
(**a**) Schematic draft of the candidalysin peptide toxin synthesized by *C. albicans* cells in hyphal morphogenic configuration [110] in (**b**) the Raman spectrum of pure candidalysin peptide. Spectroscopic differences for selected Raman signals from the candidalysin molecule (at 661, 987, and 1650 cm^−1^, as shown in inset to (**b**)) were screened upon retrieving those bands from the spectra of *C. albicans* LSEM865 in yeast and hyphae configuration (in (**c**) and (**d**), respectively). The same procedure was applied to clinical samples in the as-collected (**e**) and purified/cultured (**f**) states, respectively (i.e., from the spectra given in Figure 3a–h; cf. labels in inset). We computed the Raman spectroscopic ratio, *R_Clys_* = (*I*_661_ + *I*_987_ + *I*_1650_)/(*I*_661_ + *I*_987_ + *I*_1650_)*_ref_*, from the areas subtended by the selected Raman signals, the subscript *ref* referring to sub-band intensities in the spectrum of *C. albicans* LSEM865 yeast in (**c**). The spectroscopic parameter, *R_Clys_*, is an index for virulence. The higher the *R_Clys_* value, the higher the amount of candidalysin toxic molecules present in the sample. The computed *R_Clys_* ratio for each clinical sample is given in inset of each plot in (**e**,**f**). In the case of non-albicans clinical samples, the *R_Clys_* values were computed by using the (*I*_661_ + *I*_987_ + *I*_1650_)*_ref_* values retrieved from the average spectra of the respective standard samples. The wavenumbers indicated in inset are in cm^−1^ units and refer to key spectroscopic signals used in Raman analyses.

**Figure 13 ijms-25-11410-f013:**
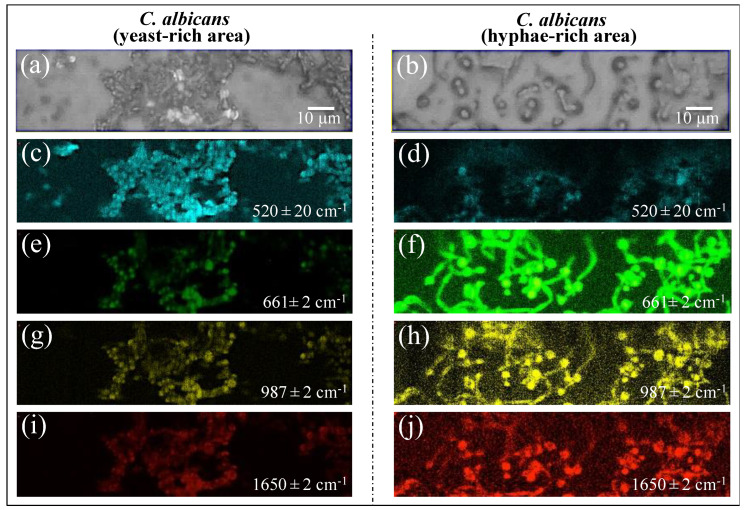
Selected zones in the as-collected samples were relatively low in biofilm and predominantly included *C. albicans* cells in yeast (**a**) and hyphal (**b**) morphology. Raman imaging was applied at spectral wavenumbers 520 ± 20, 661 ± 2, 987 ± 2, and 1650 ± 2 cm^−1^ (in (**c**/**d**), (**e**/**f**), (**g**/**h**), and (**i**/**j**), respectively), which represent sulfur chemistry (the first signal at 520 ± 20 cm^−1^) and candidalysin main signals (the latter three signals). Raman imaging gives a vivid example of how morphogenic shift into the hyphal form affects the chemical composition of cells.

**Figure 14 ijms-25-11410-f014:**
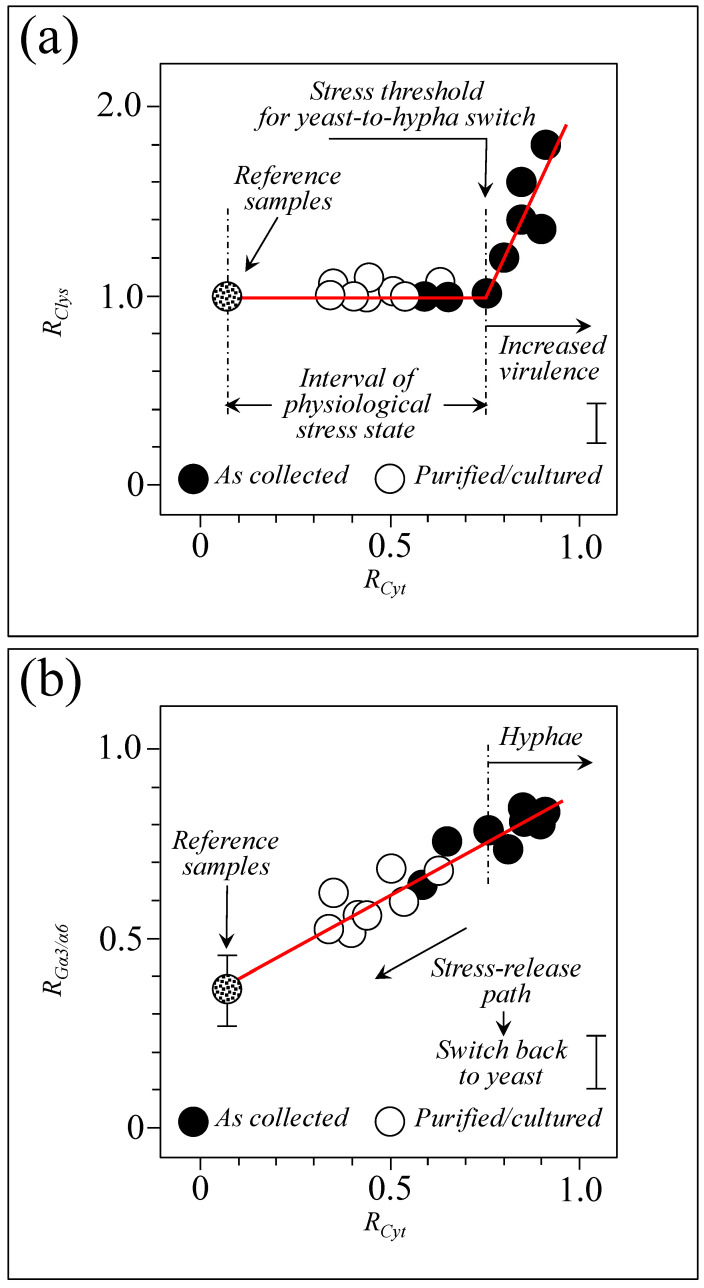
Dependences of the *R_Clys_* (**a**) and *R_G_*_α3/α6_ (**b**) spectroscopic parameters on the *R_Cyt_* ratio, which represent the stress state of cells embedded in clinical samples. The plots hint at peculiar aspects of sample diagnostics by linking virulence and cell wall structure to the stress state of the cells before and after purification–culture procedure (cf. symbols in inset), in comparison with data from reference *Candida* clades (cf. legends in inset). Maximum standard deviations recorded for sets of 10 measurements on each clinical sample are also shown in inset. Red lines were obtained by least-square fitting of the experimental data.

**Table 1 ijms-25-11410-t001:** List of the clinical samples investigated and their respective clinical characteristics; the index 1+, 2+, and 3+ in brackets stand for trace, few, and moderate, respectively.

	*Candida* Species *	*Candida* CFU/mL	Age (y)	Sex	Number of Teeth	Additional Bacteria Species
Sample 1	*C. albicans*	1 × 10^4~5^	79	Male	20	*α-Streptococcus* (3+) *Neisseria* sp. (1+)
Sample 2	*C. albicans* *C. parapsilosis* ^#^	1 × 10^5^	79	Female	8	*α-Streptococcus* (3+)
Sample 3	*C. glabrata* *C. albicans* ^#^	1 × 10^8^	80	Male	3	*α-Streptococcus* (3+) *Corynebacterium* sp. (2+)
Sample 4	*C. albicans* Filamentous fungi	1 × 10^4~5^	39	Female	12	*α-Streptococcus* (3+) *Neisseria* sp. (1+)
Sample 5	*C. parapsilosis*	1 × 10^7^	92	Female	16	*S. aureus MSSA* (3+) *α-Streptococcus* (3+)
Sample 6	*C. glabrata*	1 × 10^5^	79	Female	0	*α-Streptococcus* (2+)
Sample 7	*C. albicans* *C. glabrata* ^#^ Filamentous fungi	1 × 10^8^	81	Female	6	*α-Streptococcus* (3+)
Sample 8	*C. tropicalis* *C. albicans* ^#^ Filamentous fungi	1 × 10^7^	49	Male	26	*γ-Streptococcus* (3+) *α-Streptococcus* (2+)

* According to CHROMagar^TM^ test and microscopy analysis. ^#^ Minor fraction.

## Data Availability

Data is contained in the article.

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
