# Peer review of "Raman Spectroscopic Algorithms for Assessing Virulence in Oral Candidiasis: The Fight-or-Flight Response"

_ijms, 2024, doi:10.3390/ijms252111410_

Round 1

Reviewer 1 Report

Comments and Suggestions for Authors

This paper is a very well written report on a comprehensive study of Raman spectroscopy and imaging of oral fungi. The authors explored a range of possible markers and despite some setbacks eventually found a useful relationship that could have diagnostic value and that is relevant because it relates to pathogenicity. 

Only minor revisions are needed.

1. Please include a brief statement about the limitations of the study not already mentioned and point out directions for future work, e.g. (1) comparing the 3 Raman parameters, RGα3/α6, RCyt, and RClys, before and after various treatments (such as topical or systemic antifungal medicines, biocides etc), (2) comparing the 3 Raman parameters for C. albicans from health subjects with no oral fungal disease, and (3) comparing local microenvironments, e.g. fungi from dentures/appliances versus from the tongue or other oral mucosal sites of infection.

2. In the first paragraph of the introduction, please revise this sentence as follows: "In contrast, standard Candida strains are isolated and long-term cultured without exposure to external factors or stessors, so they exist in a conspicuously stress-free state."

3. In table 2 for sample 4 please correct the spelling from C alhicans to C. albicans.

4. In the references, there are superfluous numbers at the start of each reference that can be removed.

Reviewer 2 Report

Comments and Suggestions for Authors

The article “Raman spectroscopic algorithms for assessing virulence in oral candidiasis: The fight-or-flight response” is interesting. I have some considerations to make in order to improve the manuscript and attract its reading

- Title is not very appropriate for a research article

- Concisely outline the objectives and conclusion

- The manuscript should be restructured, differentiating between material and method, results and discussion. The discussion is not consistent with the results section. Results and discussion are mixed.

- There are paragraphs that can be eliminated, for example some related to reference 7. It is not necessary to explain findings from previous research. - The authors should control self-citations, for example 7, 15, 34….

- The 2nd and 3rd paragraph of point 4.1 is a major limitation of the study.

- Where do the 8 patients with candidiasis come from? Do you have the permission of an Ethics Committee?

Thank you very much

Round 2

Reviewer 2 Report

Comments and Suggestions for Authors

The answers are partially relevant.

Thank you very much